# SELF-ORGANIZING VISUAL EMBEDDINGS: NON-PARAMETRIC SELF-SUPERVISED LEARNING

## ABSTRACT

We present Self-Organizing Visual Embeddings (SOVE), a new training technique for unsupervised representation learning. SOVE avoids learning prototypes from scratch and instead explores relationships between visual embeddings in a non-parametric space. Unlike existing clustering-based techniques that employ a single prototype to encode all the relevant features of a concept, we propose the SOVE method, where a concept is represented by many semantically similar representations, or judges, each containing a complementary set of features that together can fully characterize the concept, and maximize training performance. We reaffirm the feasibility of non-parametric self-supervised learning (SSL) by introducing novel non-parametric adaptations of two loss functions with the SOVE technique: (1) non-parametric cluster assignment prediction for class-level representations and (2) non-parametric Masked Image Modeling (MIM) for patch-level reconstruction. SOVE achieves state-of-the-art performance on many image retrieval benchmarks. Additionally, SOVE demonstrates enhanced scaling performance when trained with Vision Transformers (ViTs), showing increased gains as more complex encoders are utilized.

## 1 INTRODUCTION

In recent years, self-supervised learning (SSL) has significantly changed how large deep learning models are trained in industry and academia. Today, most complex learning systems in vision and natural language processing (NLP) all follow a similar training strategy composed of two distinct stages: (1) a longer round of self-supervised pre-training followed by (2) a round of supervised fine-tuning on a task of interest. This strategy not only produces a robust predictive model but also reduces costs associated with data labeling. In computer vision, SSL methods (Chen et al., 2021; Silva & Ramírez Rivera, 2023; Chen et al., 2020a) based on multiview and joint-embedding architectures are effective techniques for learning representations from unlabeled images.

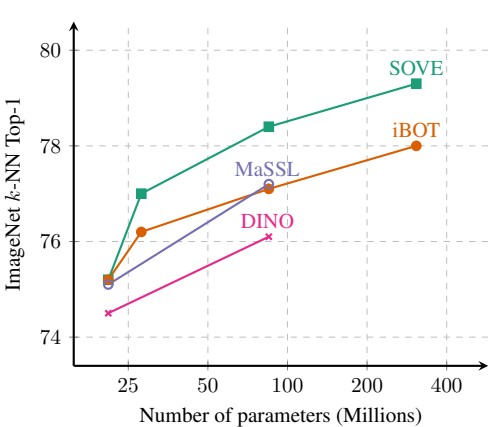

Figure 1: $k$-**NN top-1 accuracy on ImageNet.**

Current state-of-the-art SSL methods (Zhou et al., 2022; Oquab et al., 2023) follow a nearly identical framework: they learn a set of anchors (or prototypes) that are presumed to represent hidden concepts[1] in the data. Upon receiving a pair of views as input, the self-supervised training assumes that the two views should produce similar prediction patterns w.r.t. the anchors. However, when views are too dissimilar (due to extensive augmentations), the views and anchors will exhibit inconsistent prediction patterns. In this scenario, the neural network is forced to compress (discard) the unique information in each view in favor of the shared features that (1) pull the views' representations together and (2) provide the learning signal to update the anchors. In this learning framework, the

---

[1]We use the term "concept" as a generalized idea of a cluster, i.e., images that share a subset of features.

anchor repels or attracts new images to a concept. Therefore, if an anchor has a limited set of features to represent its underlying concept, the network will be compelled to discard excessive information from the views to optimize its objective, potentially harming the learned representations.

To address this limitation, we propose a non-parametric approach where *multiple* anchors represent a concept. We refer to this new method as Self-Organizing Visual Embeddings (SOVE), cf. Figure 2 (right and middle-bottom) for an initial visual description. Intuitively, each hidden concept in the data, for instance, the concept of four-legged animals or vehicles with wheels, will be represented by *multiple* anchors. Each anchor (within a concept) acts as a judge and produces a vote pertaining to the class membership of a view to that concept. However, each additional anchor produces a vote (similarity scores) against views based on distinct feature sets, supplementing the relationship criterion between views and concepts. Then, we obtain a final score for a view as a weighted combination of the individual scores from each judge within a concept. To ensure that judges within a concept share semantic characteristics, we propose a judge selection algorithm over a pool of non-parametric representations of previously seen images during training.

In contrast to existing solutions (Caron et al., 2020) where a *single* anchor represents an entire concept in the data, our proposal smooths the similarity optimization between views and anchors, by enriching the feature set of hidden concepts, allowing for consistent predictions between views. To ensure our learned representations perform well on a variety of downstream tasks, including classification and dense prediction, we extend the non-parametric SOVE algorithm to perform Masked Image Modeling (MIM), where masked representations must agree with corresponding non-masked embeddings from the perspective of multiple local-level judges.

Our contributions are threefold:

- We present the novel Self-Organizing Visual Embeddings framework to improve SSL clustering-based methods. We propose to optimize views based on the soft similarity viewpoint of a group of semantically similar embeddings that represent a given hidden concept in the space of non-parametric representations. By enriching the feature set of concepts with multiple judges, we create more complex interactions between views and concepts, preventing the neural network from discarding excessive information when optimizing for consistency.
- We demonstrate the adaptability of the MIM pretext task to a non-parametric design using the SOVE framework. The non-parametric MIM task learns fine-grained features by reconstructing local-level masked embeddings based on a non-parametric tokenizer that uses patch-level representations from different images as anchor points. This adaptation increases the performance of the learned representations and demonstrates superior performance compared to existing approaches on downstream dense prediction tasks.
- Our work demonstrates the feasibility of non-parametric clustering-based methods, where we avoid learning prototypes from random weights, and show that such an approach is stable, does not require extra regularizers to avoid mode collapse, is extensible to many pretext tasks such as MIM, and produces transferable representations. Moreover, we show that SOVE's performance increases as we scale the model architecture.

## 2 METHODOLOGY

To introduce our method, we will start with an illustrative example. Assume a concept that represents four-legged animals like cats and cows. Let $\boldsymbol{F} = \{\boldsymbol{f}_1, \boldsymbol{f}_2, \boldsymbol{f}_3, \boldsymbol{f}_4\}$, define a set of essential features for the concept, such as the animal's shape, $\boldsymbol{f}_1$, background, $\boldsymbol{f}_2$, eyes, $\boldsymbol{f}_3$, and fur texture, $\boldsymbol{f}_4$. Given two representations of an input image $\boldsymbol{z}^1$ and $\boldsymbol{z}^2$ where the first has high response to features $\{\boldsymbol{f}_1, \boldsymbol{f}_3, \boldsymbol{f}_4\}$ and the second to features $\{\boldsymbol{f}_2, \boldsymbol{f}_4\}$, only feature $\boldsymbol{f}_4$ is common among the two views. Consider two scenarios, one where the concept is poorly represented by an anchor $\boldsymbol{a}_i$ responding to features $\{\boldsymbol{f}_2, \boldsymbol{f}_4\}$ and a second where $\boldsymbol{a}_i$ responds to all features in $\boldsymbol{F}$. To optimize for consistency in the first case, i.e., to approximate the two views in the embedding space, the neural network may compress (discard) the unique features from each view, i.e., $\{\boldsymbol{f}_1, \boldsymbol{f}_2, \boldsymbol{f}_3\}$, so that $s\left(\boldsymbol{z}^1, \boldsymbol{a}_i\right) \approx s\left(\boldsymbol{z}^2, \boldsymbol{a}_i\right)$, where $s(\cdot, \cdot)$ is a similarity function such as the cosine similarity. Feature $\boldsymbol{f}_4$ will be the only factor used to propagate signals for updating the embeddings and the anchors. In the second scenario, however, the increased redundancy will allow for more complex interactions between views and anchors, and less important features will be discarded, strengthening the learning signal and enriching the learned representations.

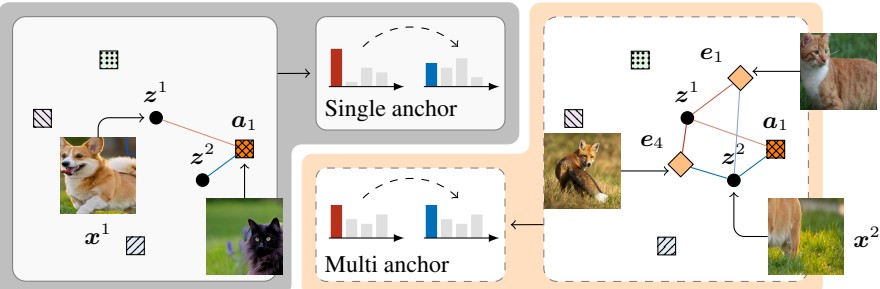

Figure 2: **Existing solutions (parametric single anchor) vs. SOVE (non-parametric multi anchors).** Clustering-based SSL methods (left) learn a finite set of anchors (colored squares) to represent hidden concepts in the data. They assume that multiple views of an image, $z^1$ and $z^2$, should agree (produce similar predictions) w.r.t. the learned anchors. However, when views are too dissimilar, their interaction with the anchors produces inconsistent prediction patterns (top-middle). To achieve consistency, the optimization process discards unique features in the views in favor of the limited shared ones, which may limit the learning of the anchors. Instead of a *single* learnable anchor, we propose having *many* non-parametric judges per concept (right). Each judge produces an individual score measuring the view's membership to the concept. Then, judges combine their votes to produce a final score. In this example, the anchor $a_1$ selects two additional judges, $e_1$ and $e_4$, to represent its concept. In this way, we can increase the agreement between views and concepts, leading to better transferable representations.

**Notation.** Let $X$ be an image dataset and $x \sim X$ a uniformly random observation. We denote by $x^v$ the $v$-th augmented version of $x$, referred to as a view of $x$, where the superscript $v$ indexes the views $V$. To create views, we use a random transformation function $t$ such that $x^v = t(x)$. For simplicity, we consider the case where $V = 2$. However, we explore multiple view scenarios in the main experiments. We denote by $f_\Phi$ a Vision Transformer (ViT) (Dosovitskiy et al., 2020) encoder with parameters $\Phi$ that receives a view and produces a matrix of representation vectors $Z^v = f_\Phi(x^v) \in \mathbb{R}^{L \times d}$, where $L$ and $d$ are the number of patch tokens and feature dimensionality respectively, such that $Z^v = \{z_l\}_{l=0}^L$ contains patch representations where the first element $z_0 \in \mathbb{R}^d$ is the classification or $\texttt{[CLS]}$ token embedding and the remaining $Z_{1:L,:}$ elements are patch embeddings from an image $x$. Instead of learning class and patch level discrete features (or prototypes) as previous work did (Caron et al., 2020; Zhou et al., 2022), we define feature sets $E_C \in \mathbb{R}^{N_C \times d}$ and $E_P \in \mathbb{R}^{N_p \times d}$ to hold $\texttt{[CLS]}$ and patch embeddings from previous iterations. Each set holds a subset of the training data features for global and local representations.

## 2.1 LEARNING REPRESENTATIONS USING SSL CLUSTERING STRATEGIES

Current SSL methods (Oquab et al., 2023; Zhou et al., 2022) have a nearly identical framework composed of two pretext tasks: (i) cluster assignment prediction over class-level embeddings and (ii) token-level embedding reconstruction or Masked Image Modeling (MIM). Usually, each task is learned with a different set of trainable parameters.

The cluster assignment prediction task aims to learn embeddings that covary w.r.t. a set of *learnable* prototypes, or anchors $\theta \in \mathbb{R}^{K \times D}$. The optimization follows:

$$\mathcal{L}_{\texttt{[CLS]},\theta} = - \sum_{x \sim X} P_\theta^{\texttt{[CLS]}}(z_0^1)^T \log\left(P_\theta^{\texttt{[CLS]}}(z_0^2)\right), \tag{1}$$

where $P_\theta^{\texttt{[CLS]}}(u) = \sigma(\langle u, \theta^T \rangle)$, $\sigma(\cdot)$ is the softmax function and $\langle \cdot, \cdot \rangle$ is the cosine similarity. Elementally, $P_\theta^{\texttt{[CLS]}}(\cdot)$ is a linear layer, parameterized by $\theta$, that maps the views' vector embeddings $z^v$ into $K$ pseudo-categories assigning each representation a *soft* distribution (prediction pattern) describing its membership probabilities to all prototypes.

This objective can be viewed from a pseudo-clustering perspective, where each anchor acts as an individual judge representing a pseudo-class, i.e., a hidden concept in the data. In both parametric and non-parametric approaches, each judge is responsible for issuing a similarity score (a vote) that relates a view to the concept it represents.

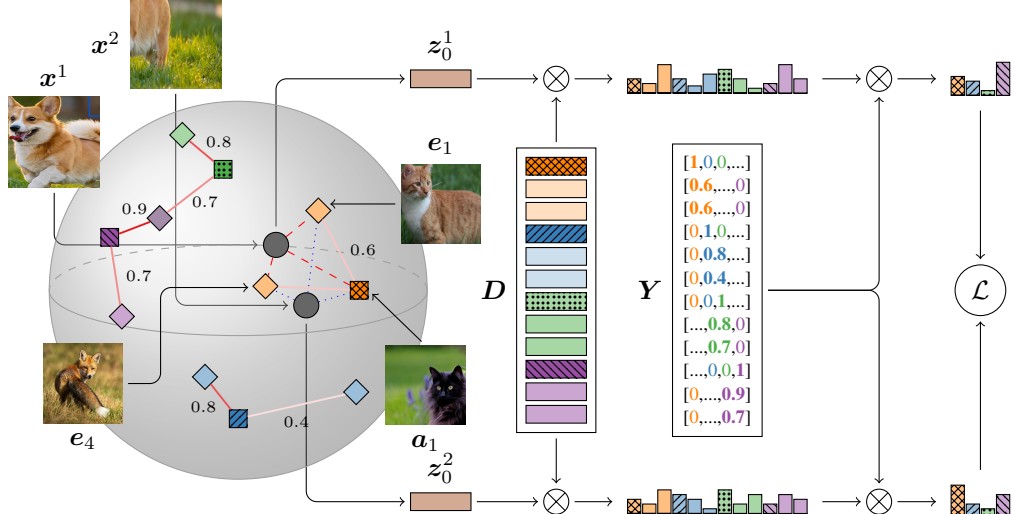

Figure 3: **Overview of the SOVE algorithm**. First, we select a set of random anchors $\boldsymbol{A} = \{\boldsymbol{a}_i\}_{i=0}^{K}$ (colored squares with patterns) from a set of representations kept in memory (gray sphere). Second, each anchor selects $k$ nearest neighbors (2 in this illustration). The selected embeddings are treated as members of the concept represented by each anchor $\boldsymbol{a}_i$, forming a dataset $\boldsymbol{D} \in \mathbb{R}^{K(k+1) \times d}$. Note that a given embedding may belong to more than one class simultaneously. We use the similarity score between the nearest embeddings and anchors as class membership confidence scores to build soft labels $\boldsymbol{Y} \in \mathbb{R}^{K(k+1) \times K}$. Intuitively, each concept contains a set of judges that independently estimate the degree of similarity between a view and a concept. Then, judges within a concept combine their votes to produce a final score for each view. Finally, the resulting similarity distribution for each view is optimized to be consistent across concepts.

Existing solutions typically employ a single judge (usually learnable) to determine the membership of a view to a concept. However, a significant challenge with this approach arises when the views are relatively different and exhibit low feature sharing, which is common due to the stochastic nature of the view generation process. In such cases, the views receive inconsistent similarity scores w.r.t. the anchors. In other words, each judge will provide inconsistent evaluations for views of the same image. We argue that this inconsistency makes the objective in (1) inefficient, leading to suboptimal downstream performance of the learned features. In the parametric case, if the two views do not share enough features, they will receive different membership scores from a judge (anchor), limiting the learning process (update of the anchors) due to a lack of redundant information between views. A similar argument applies to the non-parametric case.

This framework places excessive importance on the anchors' representations, which need to encode a comprehensive set of relevant features to fully represent their concepts. For example, in Figure 3, views $\boldsymbol{x}_1$ and $\boldsymbol{x}_2$ have inconsistent predictions with respect to anchor $\boldsymbol{a}_1$ because $\boldsymbol{x}_2$ and $\boldsymbol{a}_1$ share background features that are absent in $\boldsymbol{x}_1$ due to extensive random cropping. To address this inconsistency, the neural network could compress the grass features present in $\boldsymbol{x}_2$'s representation $\boldsymbol{z}_2$, making the embeddings $\boldsymbol{z}_1$ and $\boldsymbol{z}_2$ more similar and their relationship with $\boldsymbol{a}_1$ more consistent. However, this approach assumes that $\boldsymbol{a}_1$ fully represents the concept in terms of features. When $\boldsymbol{a}_1$ underrepresents the concept, additional features are necessary to disambiguate the relationship between views. We argue that such additional features can be found in the vicinity of $\boldsymbol{a}_1$. For instance, even though view $\boldsymbol{x}_1$ and anchor $\boldsymbol{a}_1$ have low similarity, $\boldsymbol{x}_1$ has high similarity with one of $\boldsymbol{a}_1$'s neighbors, $\boldsymbol{e}_1$, due to the resemblance between the dog and cat's fur. Additionally, both views are similar to another neighbor, $\boldsymbol{e}_4$ (red fox), which shares background information with $\boldsymbol{x}_2$ and object shape and color with $\boldsymbol{x}_1$. By treating neighbors $\boldsymbol{e}_1$ and $\boldsymbol{e}_4$ as additional representatives of $\boldsymbol{a}_1$'s concept, we enrich the feature set used to describe the concept, allowing for a more consistent matching with views. Since prediction patterns between views and concepts are obtained from the perspective of multiple observers within a concept, the neural network can avoid discarding useful features to force consistency and optimize the objective.

Motivated by this example, we present a novel training framework that leverages *multiple* anchors to enrich the feature set representing a concept, allowing for an improved agreement between views.

## 2.2 SELF-ORGANIZING VISUAL EMBEDDINGS

The inconsistency problem previously described occurs when views of the same image produce unstable associations with anchors. If a view contains unique features (not shared with the other view), these features will produce unique correlations with anchors, resulting in inconsistent prediction patterns between views. One way to solve this problem is to increase the information redundancy of a concept by augmenting its feature set.

Grounded on these ideas, we propose a multi-anchor strategy where many semantically similar anchors independently vote to measure the relationship between views and concepts. Ideally, each additional judge brings a different perspective about the concept to which it belongs, such as new essential features that represent the concept but are not either present or sufficiently strengthened in a sole anchor. In practice, this strategy induces a smoothing effect in the anchor/view relationship so that, on average, the prediction scores of each view w.r.t. the concepts (each represented by multiple anchors) are more consistent, which in turn prevents the neural network from discarding important features from the views to enforce consistency.

### 2.2.1 UNSUPERVISED NON-PARAMETRIC JUDGE SELECTION

One important consideration is how to bootstrap additional judges within a concept while maintaining semantics. If the judges do not share semantic characteristics, their contributions will be noisy, potentially hurting the learned features. Inspired by recent work on non-parametric SSL (Silva et al., 2024), we use a feature set $\boldsymbol{E}_{\mathrm{C}}$ to store representations from previously processed images during training and expand the idea of a simple feature store, to *store* and *sample* anchor representations.

First, we sample a subset of anchor representations $\boldsymbol{A} = \{\boldsymbol{a}_i\}_{i=0}^{K} \subset \boldsymbol{E}_{\mathrm{C}}$. Second, we perform spherical $k$-Nearest Neighbors using the anchors $\boldsymbol{A}$ as centroids such that $\boldsymbol{D} = \arg\max_e^k \left( \langle \boldsymbol{A}, \boldsymbol{E}_{\mathrm{C}}^T \rangle \right)$ where the $\arg\max_e^k$ operator returns the anchors and the set of the top-$k$ closest neighbors of each anchor. At this point, $\boldsymbol{D}$ can be viewed as a dataset containing $K$ pseudo-classes, each containing $k + 1$ observations, i.e., anchors $\boldsymbol{a}_i$, and their $k$ nearest neighbors. Note that this definition allows a given vector representation $\boldsymbol{e}_j \in \boldsymbol{E}$ to belong to more than one pseudo-class, cf. Figure 3.

To account for the uncertainty from the unsupervised $k$-NN selection, we build pseudo-labels $\boldsymbol{Y}$ for the dataset $\boldsymbol{D}$ to model the soft contributions of each additional neighbor (judge) towards the views. For instance, a naive strategy would treat each neighbor as a true class member, i.e., the pseudo-labels $\boldsymbol{Y}$ are represented as one-hot vector representations. We show in Section 3.7 that such a strategy is suboptimal, probably because of false positives from the $k$-NN selection. Instead, we propose soft labels $\boldsymbol{y} \in \boldsymbol{Y}$ (cf. Figure 3), such that the class indicator value (strength of the contribution) of each additional judge is defined as the embedding similarity score between itself and the anchors, i.e., $\langle \boldsymbol{e}_j, \boldsymbol{a}_i \rangle$ for $j \in \{0, 1, \ldots, K(k + 1)\}$ and $i \in \{0, 1, \ldots, K\}$. Thus, the weight contribution of each additional judge to the views is proportional to its class membership score towards the anchor.

Now, we can compute the probability distributions for each view as, $P^{[\mathrm{CLS}]}(\boldsymbol{u}) = \sigma(\langle \boldsymbol{u}, \boldsymbol{D}^T \rangle)\boldsymbol{Y}$, where $\boldsymbol{Y}$ are soft labels that sum up to one encoding the contributions of each additional judge. Compared to prototype-based losses (1), our approach swaps the learnable anchors $\theta$ by non-parametric embeddings $\boldsymbol{E}_{\mathrm{C}}$, and introduces the pseudo-labels $\boldsymbol{Y}$. Note that the matrix multiplication between the probability distribution $\sigma(\cdot)$ and the pseudo-labels $\boldsymbol{Y}$, represents the weighted combination of the judges' votes within each concept, cf. Figure 3.

Finally, we minimize the non-parametric version of $\mathcal{L}_{[\mathrm{CLS}],\theta}$ (1), as

$$\mathcal{L}_{[\mathrm{CLS}]} = -\sum_{\boldsymbol{x} \sim \boldsymbol{X}} P^{[\mathrm{CLS}]}(\boldsymbol{z}_0^1)^T \log\left(P^{[\mathrm{CLS}]}(\boldsymbol{z}_0^2)\right). \tag{2}$$

## 2.3 NON-PARAMETRIC MIM

The MIM task has been extensively explored by Zhou et al. (2022) and Oquab et al. (2023) from the parametric perspective. The task aims to produce consistent predictions between reconstructed patch embeddings and their corresponding uncorrupted representations w.r.t. a set of learnable discrete local-level features. The goal is to train an *online* local-level tokenizer $\phi$ by randomly masking a portion of the patch token representations $\boldsymbol{x} = \{\boldsymbol{x}_l\}_{l=0}^{L}$ using a binary mask $\boldsymbol{m} \in \{0,1\}^L$ such that $\hat{\boldsymbol{x}} = \{\hat{\boldsymbol{x}}_i \mid (1-m_i)\boldsymbol{x}_i + m_i\boldsymbol{e}_{\texttt{[MASK]}}\}^L$ is a corrupted version of the input image $\boldsymbol{x}$, and $\boldsymbol{e}_{\texttt{MASK}}$ is a learnable token. The corrupted input $\hat{\boldsymbol{x}}$ is fed to the encoder $\hat{\boldsymbol{Z}} = f(\hat{\boldsymbol{x}})$ and reconstructed from the uncorrupted version following:

$$\mathcal{L}_{\text{patch},\phi} = -\sum_{l=1}^{L} m_l P_\phi^{\text{patch}}(\boldsymbol{z}_l^1)^T \log\left(P_\phi^{\text{patch}}(\hat{\boldsymbol{z}}_l^1)\right), \tag{3}$$

where, similar to $\mathcal{L}_{\texttt{[CLS]},\theta}$ (1), $P_\phi^{\text{patch}}(\cdot)$ is a linear layer that computes the probability distributions w.r.t. learnable discrete features $\phi$ by soft assigning the patch tokens to $\dot{K}$ distinct discretized representations. Note that the loss $\mathcal{L}_{\text{patch},\phi}$ (3) skips the $\texttt{[CLS]}$ token $\boldsymbol{x}_0$, and optimizes different versions of the *same* image view, where one is masked.

We propose a new version of the MIM pretext task based on a non-parametric strategy. Instead of learning a set of discrete features (online tokenizer), we obtain the probability distributions $P^{\text{patch}}(\cdot)$ by exploring relationships between semantically similar patch embeddings in the space of non-parametric representations using the SOVE strategy, cf. Section 2.2.

We start by randomly sampling $\dot{K}$ anchor patch discrete tokens $\dot{\boldsymbol{A}} = \{\dot{\boldsymbol{a}}_j\}_{j=0}^{\dot{K}} \subset \boldsymbol{E}_{\text{P}}$. Then, each anchor selects $k$ nearest patch token representations to become members of a local concept represented by anchor $\dot{\boldsymbol{a}}_i$ such that, $\dot{\boldsymbol{D}} = \arg\max_e^k\left(\left\langle \dot{\boldsymbol{A}}, \boldsymbol{E}_{\text{P}}^T \right\rangle\right)$ is the dataset containing local anchors and their neighbors. Note that the $\boldsymbol{E}_{\text{P}}$ can be seen as a non-parametric or *offline* tokenizer.

Similarly to Section 2.2, we obtain the patch-level probability distributions, in a non-parametric form, as $P^{\text{patch}}(\boldsymbol{v}) = \sigma\left(\left\langle \boldsymbol{v}, \dot{\boldsymbol{D}}^T \right\rangle\right)\dot{\boldsymbol{Y}}$, and optimize

$$\mathcal{L}_{\text{patch}} = -\sum_{l=1}^{L} m_l P^{\text{patch}}(\boldsymbol{z}_l^1)^T \log\left(P^{\text{patch}}(\hat{\boldsymbol{z}}_l^1)\right), \tag{4}$$

where we remove the learnable discrete tokens $\phi$ in favor of non-parametric embeddings $\boldsymbol{E}_{\text{P}}$ and introduce the judges' soft contributions through $\dot{\boldsymbol{Y}}$.

The non-parametric MIM objective (4) encourages the network to reconstruct the missing patches so that the prediction patterns between reconstructed and original embeddings are consistent from the point of view of multiple judges within each concept.

The final loss is a convex combination of the two losses, $\mathcal{L}_{\text{SOVE}} = \lambda_1 \mathcal{L}_{\texttt{[CLS]}} + \lambda_2 \mathcal{L}_{\text{patch}}$. By default, $\lambda_1 = \lambda_2 = 1$.

## 3 MAIN EXPERIMENTS

We begin by assessing the quality of the pre-trained representations on a range downstream tasks, adhering the experimental protocol outlined by Zhou et al. (2022). Subsequently, we justify the choices in our architecture by ablating its main components.

### 3.1 LINEAR EVALUATION ON IMAGENET

$k$-**NN and Linear probing.** In Table 1, we evaluate the linear transferability power of the representations learned by SOVE using two methods: (1) non-parametric $k$-NN and (2) linear models. For the $k$-NN estimator, we sweep different values of $k$ and report the best. For linear probing, we use the pre-trained SOVE encoder as a feature extractor and train a linear layer on top of the frozen

Table 1: **Linear probing, semi-supervised fine-tuning, and $k$-NN evaluations on ImageNet-1M.**

| Method | Arch | Ep. | Lin. | 1% | 10% | $k$-NN |
|---|---|---|---|---|---|---|
| EsViT | Swin-T/14 | 300 | 78.7 | | | 77.0 |
| iBOT | Swin-T/14 | 300 | **79.3** | | | 76.2 |
| SOVE | Swin-T/14 | 300 | **79.3** | | | **77.0** |
| DeiT | ViT-S/16 | 800 | 79.8 | | | 79.3 |
| DINO | ViT-S/16 | 800 | 77.0 | 60.3 | 74.3 | 74.5 |
| iBOT | ViT-S/16 | 800 | **77.9** | **61.9** | **75.1** | 75.2 |
| MaSSL | ViT-S/16 | 800 | 77.8 | | | 75.1 |
| SOVE | ViT-S/16 | 800 | 77.8 | 61.8 | 75.0 | 75.2 |
| DeiT | ViT-B/16 | 400 | 81.8 | 75.6 | 81.4 | 81.0 |
| MoCo-v3 | ViT-B/16 | 400 | 76.7 | | | |
| NNCLR | ViT-B/16 | 1000 | 76.5 | | | |
| DINO | ViT-B/16 | 400 | 78.2 | 64.4 | 76.3 | 76.1 |
| iBOT | ViT-B/16 | 400 | 79.5 | 68.5 | 78.1 | 77.1 |
| MaSSL | ViT-B/16 | 400 | 79.6 | | | 77.2 |
| SOVE | ViT-B/16 | 400 | **79.9** | **69.5** | **78.2** | **78.4** |
| iBOT | ViT-L/16 | 250 | 81.0 | | | 78.0 |
| I-JEPA | ViT-L/16 | 600 | 77.5 | | | |
| | ViT-H/14 | 300 | 79.3 | | | |
| SOVE | ViT-L/16 | 250 | **81.2** | | | **79.2** |

Table 2: **Object detection and instance segmentation on COCO and semantic segmentation on ADE20k.** Results for ViT-B encoders.

| Method | Det. $AP^b$ | iSeg. $AP^m$ | Seg$^\dagger$ mIoU | Seg mIoU |
|---|---|---|---|---|
| Sup. | 49.8 | 43.2 | 35.4 | 46.6 |
| BEiT | 50.1 | 43.5 | 27.4 | 45.8 |
| DINO | 50.1 | 43.4 | 34.5 | 46.8 |
| iBOT | 51.2 | 44.2 | 38.3 | 50.0 |
| SOVE | **51.4** | **44.3** | **38.7** | **50.6** |

Table 3: **Transfer learning by fine-tuning SSL methods on smaller datasets.** We report top-1 accuracy for ViT-B encoders.

| Method | $C_{10}$ | $C_{100}$ | iNat$_{18}$ | iNat$_{19}$ | Flwrs | Cars |
|---|---|---|---|---|---|---|
| Rand | 99.0 | 90.8 | 73.2 | 77.7 | 98.4 | 92.1 |
| BEiT | 99.0 | 90.1 | 72.3 | 79.2 | 98.0 | 94.2 |
| DINO | 99.1 | 91.7 | 72.6 | 78.6 | 98.8 | 93.0 |
| iBOT | 99.2 | 92.2 | **74.6** | 79.6 | 98.9 | 94.3 |
| SOVE | **99.3** | **92.4** | **74.6** | **79.7** | **99.0** | **94.5** |

features. SOVE improves over existing methods by $+1.2$ top-1 accuracy on the $k$-NN benchmark. SOVE's $k$-NN top-1 accuracy (ViT-L, 307 million params) is similar to the linear top-1 accuracy of I-JEPA (Assran et al., 2023) (ViT-H, 632 million params) with a pre-training schedule of 300 epochs. Additionally, we report performance values for *supervised* baseline DeiT (Touvron et al., 2021), as well as for the strong SwinT (Liu et al., 2021) baseline EsViT (Li et al., 2022).

We observed an interesting performance scaling when training ViTs with the SOVE algorithm. As we increased the complexity of the ViT backbones, the expected performance gains were higher than those of competing methods. In Table 1, while SOVE's performance using the ViT-S backbone is similar to existing solutions, **more complex backbones, such as ViT-B/L and SwinT, produced larger performance gains**. These gains are primarily shown in the $k$-NN evaluation, suggesting a strong boost in the off-the-shelf representational power for retrieval tasks, cf. Section 3.5.

## 3.2 SEMI-SUPERVISED FINE-TUNING ON IMAGENET

In Table 1, we measure SOVE's representation capacity to learn tasks using a limited set of labeled examples. We follow the *unsupervised pre-train*, *supervised fine-tune* protocol and report top-1 accuracy using 1% and 10% of ImageNet-1M labeled images. Similar to other experiments, we observe that SOVE's performance tends to increase and surpass competing methods when trained with more complex encoders. We observe a large performance gap between SOVE and iBOT in smaller data regimes, such as with 1% labeled data. As the fraction of annotated data increases, performances tend to level out. Following previous work (Chen et al., 2020b), we fine-tune the pre-trained encoders for 1000 epochs from the first layer of the projection head.

## 3.3 DENSE PREDICTION TASKS

Dense prediction tasks involve multiple predictions per input observation. We consider three downstream evaluations on (1) object detection, (2) semantic segmentation, and (3) instance segmentation. To solve detection and segmentation tasks, the learned representation needs to encode information regarding the objects' localization and their classes. An optimal fixed-size representation needs to strike a balance between coarse and fine-grained features used to classify and segment objects by performing pixel predictions.

**Object Detection and Instance Segmentation on COCO.** In Table 2, first and second columns, we report $AP^b$ and $AP^m$ for various SSL methods on the COCO dataset using the Mask R-CNN (He

et al., 2017) as the task layer. The entire network is fine-tuned for 12 epochs, following Zhou et al.'s (2022) protocol. SOVE 's representations exhibit modest improvements of $+0.2$ in AP$^b$ and $+0.1$ in AP$^m$ over iBOT on object detection and inst. segmentation tasks, respectively.

**Semantic Segmentation on ADE20K.** In Table 2, third and fourth columns, we report mean intersection over union (mIoU) for semantic segmentation on the ADE20K dataset (Zhou et al., 2017). Following Zhou et al. (2022), we consider two protocols (1) linear probing and (2) fine-tuning. In the first, we keep the patch tokens from the pre-trained SOVE encoder fixed and only train a linear model on top of the frozen features. In the second, we use the task layer in UPerNet (Xiao et al., 2018) and finetune all the network's parameters. In both scenarios, SOVE pre-trained representations improved iBOT's strong baselines by $+0.4$ and $+0.6$ and broadened the gap to the supervised baselines by $+3.3$ and $+5.0$ mIoU, respectively.

### 3.4 TRANSFER LEARNING

In Table 3, we study transfer learning tasks using SOVE pre-trained encoders as initialization to perform fine-tuning on several classification tasks using smaller datasets. We report top-1 accuracy for six datasets including CIFAR-10/100, iNaturalist 2018/2019, Oxford 102 Flower, and Stanford Cars. SOVE encoders achieve strong downstream performances on fine-tuning protocols, surpassing competitors on **5 out of 6 datasets** with modest gains. We hypothesize that due to the long fine-tuning regime from Zhou et al.'s (2022) protocol of 1000 epochs, most methods end up reaching similar performances, also indicating saturation.

### 3.5 IMAGE RETRIEVAL

**Image retrieval.** To assess the image retrieval properties of SOVE, we consider the revisited Oxford and Paris image retrieval datasets (Radenović et al., 2018). Each dataset has three sets of increasing difficulty. We use frozen pre-trained encoders as feature extractors and apply $k$-NN on top of the frozen features. In Table 5, we report Mean Average Precision (mAP) for the Medium (M) and Hard (H) splits. SOVE significantly improves over current state-of-the-art methods, increasing mAP performance by up to $+3.2$ on the **H**ard split of both benchmarks. For reference, we report results from a supervised retrieval-specific method (Revaud et al., 2019).

**Video instance segmentation.** In Table 4, we employ frozen patch tokens from SOVE pre-trained models to perform video scene segmentation using a nearest neighbor classifier between consecutive frames. Since we do not update any extra parameters, this evaluation is particularly interesting to validate the fine-grained downstream capabilities of SOVE frozen features learned through reconstruction using the non-parametric MIM loss (4). We compare SOVE's performance to existing SSL methods and to a supervised ViT-S/8 trained on ImageNet-1M. SOVE improves upon the iBOT baseline by up to $+1.3$ on mean contour-accuracy $\mathcal{F}_m$.

### 3.6 ROBUSTNESS

We evaluate SOVE's performance on a robustness test over seven variations of foreground/background mixing and masking using the ImageNet-9 dataset (Xiao et al., 2020). We report results in Table 6 for ViT-B encoders. SOVE significantly outperforms competitors in **six out of the seven** background changes with significant gains on most of the categories such as: *Only-FG* (OF) $+2.3$, *Mixed-Rand* (MR) $+2.6$, *Mixed-Next* (MN) $+2.7$, and *Only-BG-B* (OBB) $+2.3$.

### 3.7 ABLATIONS

To understand why SOVE learns useful visual representations using unsupervised data, we explore its main components and our reasoning for choosing the optimal set of hyper-parameters.

**Online vs. non-parametric tokenizers.** In Table 7, we compare the performance of methods using online or pre-trained tokenizers with our non-parametric approach, using ViT-S encoders pre-trained for 300 epochs without multi-crop augmentation. We ablate the effect of each loss function, (2) and (4). The symbol $\Delta$ denotes methods that use a pre-trained DALL-E encoder (Ramesh et al., 2021) as a tokenizer. We observe that pre-training without the $\mathcal{L}_{\texttt{[CLS]}}$ loss (2) negatively affects

Table 4: **Video object segmentation on DAVIS 2017.** We report mean region similarity $\mathcal{J}_m$ and mean contour-based accuracy $\mathcal{F}_m$.

| Method | Data | Arch. | $(\mathcal{J}\&\mathcal{F})_m$ | $\mathcal{J}_m$ | $\mathcal{F}_m$ |
|---|---|---|---|---|---|
| *Sup.* | | | | | |
| IN-1K | IN-1K | ViT-S/8 | 66.0 | 63.9 | 68.1 |
| STM | I/D/Y | RN50 | 81.8 | 79.2 | 84.3 |
| *Self-Sup.* | | | | | |
| CT | VLOG | RN50 | 48.7 | 46.4 | 50.0 |
| MAST | YT-VOS | RN18 | 65.5 | 63.3 | 67.6 |
| STC | Kinetics | RN18 | 67.6 | 64.8 | 70.2 |
| DINO | IN-1K | ViT-S/16 | 61.8 | 60.2 | 63.4 |
| | IN-1K | ViT-B/16 | 62.3 | 60.7 | 63.9 |
| iBOT | IN-1K | ViT-S/16 | 61.8 | 60.4 | 63.2 |
| | IN-1K | ViT-B/16 | 62.7 | 61.7 | 63.7 |
| SOVE | IN-1K | ViT-B/16 | **63.3** | **61.7** | **65.0** |

Table 5: **Image retrieval.** We report mAP using *off-the-shelf* features.

| Method | Arch. | Epo. | $\mathcal{R}\mathcal{O}$x M | $\mathcal{R}\mathcal{O}$x H | $\mathcal{R}$Par M | $\mathcal{R}$Par H |
|---|---|---|---|---|---|---|
| Sup. | RN101 | 100 | 49.8 | 18.5 | 74.0 | 52.1 |
| DINO | ViT-B/16 | 400 | 37.4 | 13.7 | 63.5 | 35.6 |
| iBOT | ViT-B/16 | 400 | 36.8 | 14.3 | 64.1 | 36.6 |
| MaSSL | ViT-B/16 | 400 | 39.3 | 14.1 | 65.8 | 38.1 |
| SOVE | ViT-B/16 | 400 | **42.7** | **17.5** | **67.3** | **41.3** |

Table 6: **Robustness against background changes.** Results for ViT-B encoders.

| Method | Background Changes | | | | | | | Clean |
|---|---|---|---|---|---|---|---|---|
| | OF | MS | MR | MN | NF | OBB | OBT | IN-9 |
| iBOT | 91.9 | 89.7 | 81.9 | 79.7 | 54.7 | 17.6 | 20.4 | 96.8 |
| MaSSL | 91.0 | 90.2 | 83.0 | 80.4 | 53.4 | 15.8 | 23.7 | 97.6 |
| SOVE | **93.3** | **91.4** | **85.6** | **83.1** | **55.8** | **19.9** | 22.8 | 97.1 |

Table 7: **Parametric vs non-parametric tokenizers.** $\Delta$: pre-trained DALL-E encoder.

| Method | $\mathcal{L}_{\texttt{[MIM]}}$ | $\mathcal{L}_{\texttt{[CLS]}}$ | $k$-NN | Lin. |
|---|---|---|---|---|
| iBOT | ✓ | ✓ | 69.1 | 74.2 |
| | ✓ | ✗ | 9.5 | 29.8 |
| BEIT | $\Delta$ | ✗ | 6.9 | 23.5 |
| DINO | ✗ | ✓ | 67.9 | 72.5 |
| BEIT+DINO | $\Delta$ | ✓ | 48.0 | 62.7 |
| SOVE | ✓ | ✓ | **70.0** | **74.3** |
| | ✓ | ✗ | 16.8 | 30.2 |
| | ✗ | ✓ | 68.6 | 72.7 |

Table 8: The effect of having additional judges on each pretext task. We report top-1 $k$-NN accuracy.

| | # of Judges [CLS] | | | | |
|---|---|---|---|---|---|
| [MIM] | 1 | 3 | 5 | 7 | 9 |
| 1 | 69.2 | 69.4 | 69.3 | 69.7 | 70.2 |
| 3 | | 68.9 | 69.3 | 69.3 | 69.9 |
| 5 | | | 68.8 | 69.2 | 69.8 |
| 7 | | | | 69.2 | 69.3 |
| 9 | | | | | 69.0 |

the performance of the learned representations. However, SOVE 's non-parametric strategy (4) outperforms the parametric counterpart by 7.3% accuracy points in $k$-NN, suggesting that non-parametric MIM learns faster and contributes more to the final representation.

**On the number of additional judges.** In Table 8, we examine the impact of incorporating additional judges into each of the proposed loss functions, (2) and (4). Our findings indicate that the global loss function (2), which operates on [CLS] tokens, benefits significantly from the inclusion of more judges, demonstrating a clear trend of performance improvement as the number of judges per concept increases. Conversely, for the local level [MIM] loss (4), Table 8 shows that the addition of multiple judges does not lead to performance enhancements.

**Modeling the contributions of addition judges.** As described in Section 2.2.1, additional judges are selected as the closest embeddings to the concept's anchor using spherical $k$-NN. When combining the individual scores of

Table 9: Modeling the individual contributions of additional judges in the SOVE algorithm.

| Method | Soft | One Hot |
|---|---|---|
| $k$-NN | **70.2** | 69.8 |

judges within a concept, the contribution of each judge to a view is proportional to its distance from the anchor. Consequently, judges closer to the anchor have a stronger influence on the view membership calculation than those farther away. In Table 9, we explore an alternative approach to modeling the contributions of additional judges. Instead of using the distance to the concept's anchor as the contribution weight, we assign a one-hot distribution to each additional judge, meaning that all judges within a concept contribute equally when computing the views' membership.

SOVE demonstrates robustness to both methods, with a slight preference for our soft-contribution approach, as discussed in Section 2.2.1.

## 4 RELATED WORK

**Clustering and representation learning.** Combining clustering and deep learning has been a long-promising approach for unsupervised visual representation learning. Caron et al. (2018; 2019); Van Gansbeke et al. (2020) incorporated classic methods such as $k$-Means and $k$-NN in a deep unsupervised learning framework for visual features. Asano et al. (2020) proposed a self-labeling unsupervised method as an instance of the optimal transport problem. Caron et al. (2020) proposed a mini-batch version of the Sinkhorn-Knopp algorithm (Cuturi, 2013) to optimize cluster assignments between views of an image. Silva & Ramírez Rivera (2022) follows the clustering idea using SGD. Caron et al. (2021) scaled previous ideas to ViTs (Dosovitskiy et al., 2020). Inspired by modern NLP methods (Devlin et al., 2019), Zhou et al. (2022) investigated the masked image modeling (MIM) pre-text task, also studied by Bao et al. (2021). These methods require special regularization techniques, such as centering, sharpening, and Sinkhorn-Knopp, to avoid ill-posed states.

**Non-parametric SSL.** The term non-parametric does not imply learning systems without parameters. Instead, it describes a framework where the relationship between variables can be derived from the data without assuming any parametric form (Sanborn et al., 2024). Wu et al. (2018) proposed a non-parametric alternative to the parametric softmax classifier to solve unsupervised classification problems at the instance level using Noise Contrastive Estimation (NCE) to approximate the full softmax and a memory bank containing features from previous iterations to sample positive and negative representations following the noise distribution. Subsequent work by He et al. (2020) and Chen et al. (2021) builds upon this idea but uses augmented versions of the same image (views) as positives. He et al. (2020) employed a memory bank to sample negative pairs and optimizes a variation of the NCE loss, termed the InfoNCE (Oord et al., 2018). Chen et al. (2020a) avoided an external memory by exploring in-batch representations to sample negatives. Similarly, Dwibedi et al. (2021) optimized the InfoNCE using different images as positive pairs. For each input image, the most similar representation in memory is taken as a positive and the rest of the representations in memory are deemed as negatives. Recently, Silva et al. (2024) proposed a non-parametric approach for clustering-based SSL. The main learning assumption is that views of an image should produce similar prediction patterns when compared to previous concepts stored in memory.

**SOVE.** Different from previous approaches, SOVE learns image embeddings by taking into consideration the viewpoint of many semantically similar anchors (judges) from different images representing a hidden concept in the data. Each judge encodes different aspects of a concept to enrich the concept's features avoiding excessive compression of important features. SOVE does not require negative sampling and does not optimize the NCE or the InfoNCE objectives. SOVE is a general framework, and under a strict configuration, it is equivalent to the framework of Silva et al. (2024). In addition, SOVE proposes the novel non-parametric MIM loss, where the reconstruction task is based on the viewpoint of local-level embeddings from different images in a non-parametric space.

## 5 CONCLUSIONS

We presented Self-Organizing Visual Embeddings, a novel SSL pre-training strategy to learn effective representations from unlabeled images. SOVE addresses the problem of underrepresented concepts in clustering-based SSL methods, by enhancing the feature set of concepts through multiple anchors that live in a semantically meaningful region in the non-parametric space of features. SOVE avoids leaning prototypes and presents two novel non-parametric pre-text tasks that are stable to train and do not require extra regularizations to avoid ill-posed solutions. Our comprehensive benchmarking shows that SOVE's visual representations are state-of-the-art in many downstream tasks such as object detection, instance and semantic segmentation, image retrieval, and linear probing. Additional improvements such as hyper-parameter tuning, extra regularizers, and scaling techniques, as studied by Oquab et al. (2023), can potentially improve SOVE's performance and are reserved for future work.

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

## A    Implementation Details

For the main experiments in Section 3, we train SOVE using three Vision Transformer architectures: ViT-Small, ViT-Base, and ViT-Large, with 21, 85, and 307 billion parameters, respectively. In addition to the classic ViT architecture, we train SOVE using a Swin-Transformer backbone containing 28 billion parameters. Following previous methods (Caron et al., 2021; Zhou et al., 2022), we create twelve views of the same image at each training iteration. Views indexed from $v = \{0, 1\}$ have shape $x^v \in \mathbb{R}^{224 \times 224 \times 3}$, and views indexed from $v = \{2, 3, 4, ..., 11\}$ have shape $x^v \in \mathbb{R}^{96 \times 96 \times 3}$.

The feature sets $E_C \in \mathbb{R}^{N_C \times d}$ and $E_p \in \mathbb{R}^{N_p \times d}$ hold vector representations from global and local patches, respectively. In practice, we use $N_C = 65536$, $N_p = 8192$, and set the feature dimensionality to $d = 256$. At each iteration, we update the two feature sets following a FIFO (first-in first-out) strategy. For $E_C$, we select the `[CLS]` token representation from one of the global views and insert it into one end of $E_C$. For $E_p$, we randomly pick one of the local patch embeddings using a uniform distribution and insert it into one end of $E_p$.

For the unsupervised non-parametric judge selection algorithm (2), we *uniformly* sample $K = 8192$ anchors. Each anchor selects an additional $k = 8$ neighbors, resulting in a total of 9 judges per concept. Refer to Section 3.7 for additional context on the optimal number of judges. After judge selection, we create the pseudo-dataset $\boldsymbol{D} \in \mathbb{R}^{K(k+1) \times d}$, where $K$ is the number of anchors, $k$ is the number of additional judges, and $d = 256$ is the feature vector dimensionality. Likewise, the pseudo-labels are $\boldsymbol{Y} \in \mathbb{R}^{K(k+1) \times K}$.

For the non-parametric MIM loss (4), we sample $\dot{K} = 512$ anchors. As shown in Table 8, the non-parametric MIM loss does not seem to benefit from multiple judges. Thus, we use a single judge, the anchor itself, to represent a local concept. Consequently, the pseudo-dataset and labels have shapes $\dot{\boldsymbol{D}} \in \mathbb{R}^{\dot{K} \times d}$ and $\dot{\boldsymbol{Y}} \in \mathbb{R}^{\dot{K} \times \dot{K}}$.

In practice, for the global loss (2), given the values of $K = 8192$ and $k = 8$, the pseudo-dataset $\boldsymbol{D}$ has shape $\mathbb{R}^{73728 \times 256}$. Likewise, for the non-parametric MIM local loss (4), with $\dot{K} = 512$, the pseudo-dataset $\dot{\boldsymbol{D}}$ has shape $\mathbb{R}^{512 \times 256}$.

## B    Extended Experiments

### B.1    Time and Computing Trade-off

In Table B.1, we present the trade-off between parametric and non-parametric SSL. Following the exact protocol from (Silva et al., 2024), we report the training time and memory requirements for SOVE and existing solutions. The main difference between iBOT/DINO and SOVE is the absence of learnable prototypes in SOVE. Instead, SOVE employs two feature sets, $\boldsymbol{E}_C$ and $\boldsymbol{E}_p$, to store `[CLS]` and patch-level representations, respectively. In contrast, iBOT learns two separate sets of prototypes: one for `[CLS]` tokens and a second for patch-level tokens trained with MIM. From a resource perspective, learning the prototypes requires extra memory to store gradients for updating the prototypes during the backward pass. SOVE on the other hand, updates the prototypes following a simpler FIFO strategy. Despite this, the overall training time and memory requirements for pre-training SOVE on ImageNet-1M are very similar to those of iBOT.

Table B.1: Training time and memory: We report top-1 $k$-NN performance on ImageNet-1M (accuracy), training time (hours), and memory (gigabytes) for SSL methods using ViT-S/16 backbones.

| | 100 epochs | | 300 epochs | | 800 epochs | | |
|---|---|---|---|---|---|---|---|
| | $k$-NN | Time | $k$-NN | Time | $k$-NN | Time | Mem |
| DINO | 69.7 | 24.2h | 72.8 | 72.6h | 74.5 | 180.0h | 15.4GB |
| iBOT | 71.5 | 24.3h | 74.6 | 73.3h | 75.2 | 193.4h | 19.5GB |
| MaSSL | 72.7 | 24.2h | 74.7 | 72.4h | 75.1 | 177.3h | 15.1GB |
| SOVE | 72.8 | 24.4h | 74.7 | 73.3h | 75.2 | 193.5h | 19.4GB |

## B.2 SEMI-SUPERVISED EVALUATIONS WITH FROZEN FEATURES

In Table 1, we assessed the semi-supervised performance of SSL methods using the *unsupervised pre-train* and *supervised fine-tune* paradigm. Additionally, in Table B.2, we compare the performance of multiple SSL methods on the semi-supervised learning task using frozen, *off-the-shelf* features on the ImageNet dataset. We report $k$-NN top-1 accuracy for the best-performing value of $k \in 10, 20, 100, 200$ using the data splits provided by Chen et al. (2020a).

SOVE's performance significantly improves as model complexity increases. For ViT-S backbones, SOVE performs comparably to iBOT in both data regimes. However, with the more complex ViT-B and SwinT backbones, the performance gap between SOVE and its competitors widens significantly, yielding gains of $+2.3$ and $+1.4$ for ViT-B on data regimes of 1-10% labels, respectively.

We emphasize the still large gap between supervised methods (Touvron et al., 2021) and unsupervised methods on retrieval-based tasks. Specifically, for low data regimes, the existing gap suggests that current SSL methods still have room for improvement.

Table B.2: Semi-supervised evaluations with frozen features on ImageNet-1M: We report $k$-NN top-1 accuracy using 1-10% of labels. For reference, we include results from supervised DeiT (Touvron et al., 2021).

| Method | Arch. | 1% | 10% |
|---|---|---|---|
| *Supervised* | | | |
| DeiT | ViT-S/16 | 77.3 | 78.7 |
| DeiT | ViT-B/16 | 80.2 | 80.9 |
| *Self-supervised* | | | |
| DINO | ViT-S/16 | 61.3 | 69.1 |
| | ViT-B/16 | 63.6 | 71.0 |
| iBOT | ViT-S/16 | 62.3 | 70.1 |
| | ViT-B/16 | 66.3 | 72.9 |
| | SwinT-14 | 64.2 | 71.5 |
| SOVE | ViT-S/16 | 62.2 | 70.3 |
| | ViT-B/16 | **68.6** | **74.3** |
| | SwinT-14 | 65.3 | 72.3 |

## B.3 DENSE PREDICTION TASKS

In Table B.3, we provide additional metrics for object detection, instance segmentation, and semantic segmentation evaluations using SOVE's ViT-B backbone. For object detection and instance segmentation, we use the Cascade Mask R-CNN as the task layer and the COCO dataset (Lin et al., 2014). In addition to the metrics reported in Table 2, we include $AP^b 50$ and $AP^b 75$ for object detection, and $AP^m 50$ and $AP^m 75$ for instance segmentation.

For semantic segmentation on ADE20k (Zhou et al., 2017), we follow the protocol from Zhou et al. (2022) and consider two scenarios: (1) training a linear layer on top of the frozen encoder, and (2) using UPerNet as the task layer.

Table B.3: Additional results for object detection, instance segmentation, and semantic segmentation using ViT-B encoders.

| Method | Det. & Inst. Seg. w/ Cascade Mask R-CNN | | | | | | Seg. w/ Lin. | | Seg. w/ UperNet | |
|---|---|---|---|---|---|---|---|---|---|---|
| | $AP^b$ | $AP^b_{50}$ | $AP^b_{75}$ | $AP^m$ | $AP^m_{50}$ | $AP^m_{75}$ | mIoU | mAcc | mIoU | mAcc |
| Sup. | 49.8 | 69.6 | 53.8 | 43.2 | 66.6 | 46.5 | 35.4 | 44.6 | 46.6 | 57.0 |
| DINO | 50.1 | 68.5 | 54.6 | 43.5 | 66.2 | 47.1 | 27.4 | 35.5 | 45.8 | 55.9 |
| iBOT | 51.2 | 70.8 | 55.5 | 44.2 | 67.8 | 47.7 | 38.3 | 48.0 | 50.0 | 60.3 |
| SOVE | **51.4** | **70.9** | 55.5 | **44.3** | **68.0** | **47.8** | **38.7** | **48.1** | **50.6** | **60.5** |

## C    EXTENDED ABLATIONS

**Multiple tasks improve the learned representations.**    As explained in Section 2.2, the SOVE algorithm first samples a subset of anchors $A = \{a_i\}_{i=0}^K \subset E_C$, where each anchor represents a hidden concept in the data. Then, each anchor $a_i$ further selects additional representatives (judges) through $k$-Nearest Neighbor. Hence, each concept is represented by its anchor $a_i$ and $k$ additional judges $e_j$, as per the $k$-NN criterion.

One important consideration is that this process can be done *many* times per training iteration. In Table C.1, we report the effect of such a strategy for each of the pretext tasks described in Section 2.2.1 and Section 2.3. We observe a positive trend as the number of SOVE tasks performed per training iteration increases. Moreover, Table C.1 suggests that both, global and local, tasks benefit from this strategy.

Table C.1: The effect of the number of independent pretext tasks per iteration.

| # of Tasks `[CLS]` | | | |
|---|---|---|---|
| 1 | 2 | 4 | `[MIM]` |
| 68.0 | 68.5 | 68.6 | 0 |
| 69.6 | 70.0 | 70.0 | 1 |
| | 69.7 | 69.7 | 2 |
| | | 70.2 | 4 |

**Learning global-level features: `[CLS]` vs. average patch embeddings.**    In Table C.2, we explore common strategies to learn class level representations with ViTs. We compare the (i) default strategy that uses a dedicated `[CLS]` token embedding to learn the global-level information in an image, against the (ii) alternative strategy that uses the average representation from the patch-level embeddings. For SOVE, the default strategy of using the `[CLS]` token results in a significantly more useful representations.

Table C.2: Global-level representations as `[CLS]` vs AVG. patch tokens.

| | `[CLS]` | AVG. Patch |
|---|---|---|
| $k$-NN | 69.2 | 67.8 |

Table C.3: Does the number of class-level anchors matter?

| $K$ | 1024 | 2048 | 4096 | 8192 | 16384 |
|---|---|---|---|---|---|
| $k$-NN | 68.3 | 68.9 | 69.2 | 70.2 | 69.7 |

**The masking strategy.**    In Figure C.1, we explore two masking strategies for the non-parametric MIM task: blockwise, and random masking. The blockwise algorithm follows the iterative technique proposed by Bao et al. (2021) where, at each iteration, a block of the image is randomly masked. For each block, the algorithm selects a random size (number of patches) and a random aspect ratio for the block. The algorithm repeats until the masking ratio is satisfied. The random masking strategy randomly masks patches of an image following a mask ratio. We use a masking ration of $0.3$ (30%) for blockwise masking and $0.7$ for random masking. Figure C.1 shows a consistent performance gain (top-1 $k$-NN accuracy) for the blockwise strategy.

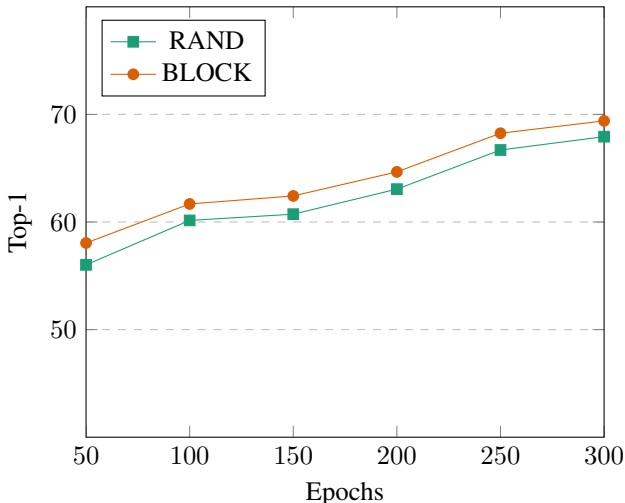

Figure C.1: Blockwise vs. Random masking.

Table C.4: Does the number of patch-level anchors matter?

| $\dot{K}$ | 256 | 512 | 1024 |
|---|---|---|---|
| $k$-NN | 69.0 | 69.2 | 69.2 |

Table C.5: The effect of the momentum hyperparameter on the teacher encoder.

| $m$ | .992→1 | .994→1 | .996→1 |
|---|---|---|---|
| $k$-NN | 69.2 | 69.4 | 69.4 |

**Does the number of `[CLS]` anchors matter?**   As described in Section 2.2.1, the first step of the SOVE algorithm is to select a subset of anchor representations from a set of embeddings from previous iterations $E_C$. These anchors intuitively represent hidden concepts in the data and are used as comparison standpoints to learn consistency between views. In Table C.3, we explore how the anchor sampling size affects SOVE's learning capabilities. The experiment suggests that as we increase the anchors' sampling size, the representations $k$-NN accuracy also increases up until 8192. Nevertheless, SOVE is very robust to the number of anchors without significant performance changes under different configurations.

**Does the number of local-level anchors matter?**   Similar to the global `[CLS]` task, to perform non-parametric MIM, we sample a subset of patch-level anchors $\dot{A}$ from a set of stored embeddings from previous iterations $E_P$. Each of these anchors represents a local concept in the embedding space where patch-level representations share semantic features. In Table C.4, we study the sampling size of local anchors and its impact on the $k$-NN performance of the learned representations. In general, SOVE shows robustness to many sampling sizes.

**Updating the momentum encoder.**   As a standard practice in SSL, SOVE employs two sibling encoders, viewed as a teacher-student setup. The student encoder receives gradient updates while the teacher encoder receives updates following a moving average from the weights of the student following, $\Phi_t = m\Phi_t + (1 - m)\Phi_s$, where $\Phi_s$ and $\Phi_t$ are the weights of the student and teacher encoders respectively. This framework can also be interpreted from a distilling perspective where the teacher distills knowledge from previous iterations into the student. In Table C.5, we study the effect of the hyper-parameter $m$ on the learned representations.

