# OpenReview forum: "Self-Organizing Visual Embeddings for Non-Parametric Self-Supervised Learning"
_ICLR.cc/2025/Conference — Submitted to ICLR 2025_

### Official Review · Reviewer_w3q7 · 2024-10-29

**Soundness:** 2
**Presentation:** 2
**Contribution:** 1
**Rating:** 3
**Confidence:** 4

**Summary:**

This paper presents Self-Organizing Visual Embeddings (SOVE), a technique for unsupervised representation learning that explores relationships between visual embeddings in a non-parametric space. SOVE uses multiple semantically similar representations, or "judges," to capture complementary features of concepts, enhancing training performance. It introduces two novel non-parametric loss function adaptations: (1) non-parametric cluster assignment prediction and (2) non-parametric Masked Image Modeling (MIM). SOVE achieves state-of-the-art performance in various benchmarks, including transfer learning and object detection, and shows improved scalability with Vision Transformers (ViTs).

**Strengths:**

1. The overall demonstration is good.
2. The proposed SSL method seems to be reasonable.

**Weaknesses:**

1. I am worried about the support for the claim of the motivation. "However, when views are too dissimilar (due to extensive augmentations), views and anchors will exhibit inconsistent prediction patterns. " Could the authors provide more clear illustration or pilot experimental results for this? This claim is not intuitive.
2. The essence of this method targets on using similar embedding for replacing the original biased feature representation (like IBoT), and the experimental results show that such mechanism could enhance the knn performance, but the fine-tuned performance merely shows marginal enhancement compared to other methods. From another perspective, this method is quite specifically adapted to KNN task? Overall, such a performance elation is not convincing regarding the effectiveness of this method.

**Questions:**

None

---

> ### Author Response · Authors · 2024-11-23
> **Support for the claim of the motivation**
>
> > I am worried about the support for the claim of the motivation. "However, when views are too dissimilar (due to extensive augmentations), views and anchors will exhibit inconsistent prediction patterns. " Could the authors provide more clear illustration or pilot experimental results for this? This claim is not intuitive.
>
> We apologize for the lack of clarity and absence of additional experimentation to back up our claims. Though we can not provide additional experimentations at this phase of the review, we may refer the reviewer to past work where augmentations in SSL have been thoroughly studied.
> As per our motivation and intuition in the paper, views need to share important features to allow the SSL mechanism to achieve good representations. Intuitively, it is a spectrum. In one end, the views could be identical crops of the original image. In this case, though the views share all features, the neural network would have an easier job of learning similar embeddings for the views. However, the learning dynamics are more difficult in this scenario, and collapse solutions may happen if not prevented properly. On the other hand, we could create views completely different from each other, with no feature sharing. This would make the SSL task more difficult and experiments show that the learned representations are not optimal, due to lack of important feature sharing. Ideally, the optimal strategy seems to be the “right” amount of augmentation where the relevant signal is preserved (shared) such as the class information and irrelevant information can be discarded in the learning process. These results are reported in [1] and [2] and discussed in [3].
>
> [1]- https://arxiv.org/abs/2002.05709
>
> [2]- https://arxiv.org/abs/2006.07733
>
> [3]- https://www.mdpi.com/1099-4300/26/3/252

---

> ### Author Response · Authors · 2024-11-23
> **Essence of the method**
>
> > The essence of this method targets on using similar embedding for replacing the original biased feature representation (like IBoT), and the experimental results show that such mechanism could enhance the knn performance, but the fine-tuned performance merely shows marginal enhancement compared to other methods. From another perspective, this method is quite specifically adapted to KNN task? Overall, such a performance elation is not convincing regarding the effectiveness of this method.
>
> We agree with the reviewer that SOVE’s best results show up in retrieval-based tasks. However, we call the reviewer’s attention to the range of evaluation methods reported in the paper. Even for the retrieval (kNN) evaluations, SOVE demonstrates strong results in different scenarios such as Table 7 where we report results on 7 distinct background changes in a robustness analysis.
>
> Moreover, Table 5 reports large performance gains over existing solutions for two datasets with increasingly difficult levels and distinct retrieval tasks.
>
> Moreover, SOVE archives superior or on-par performance on Linear probing, semi-supervised finetuning, and k-NN evaluations (Tab 1) on ImageNet-1M. We report benchmark results for object detection, instance segmentation (Tab 2), and semantic segmentation on videos (Tab 4). In all experiments, SOVE either outperforms existing methods or performs equally well.
>
> Given the variety of evaluation protocols and the performance displayed by our proposed approach, we strongly believe that the representations learned by our method are general and applicable to a large number of downstream problems.
>
> Regarding fine-tuning, we hypothesize that such benchmarks might be saturated. After all, with fine-tuning we change all the learnable weights for 1000 epochs using a supervised approach. Under such protocol, it is expected that the differences are smaller since that is pretty much a supervised learning problem under very similar setups. We follow this protocol since it is used in other papers and allows for a fair comparison.

---

### Official Review · Reviewer_S7GZ · 2024-11-01

**Soundness:** 3
**Presentation:** 1
**Contribution:** 2
**Rating:** 5
**Confidence:** 4

**Summary:**

This paper proposes an alternative objective function and methodology for enabling multi-anchor representation learning for self-supervision. The authors propose a similarity based method to sample additional anchors, randomly sampling per iteration a set of anchors from previous iterations representations, incorporating additional views or judges from a k-nn of the representation of the anchors, then producing a new probability distribution by scaling this via the original views of the object. This produces a probability distribution for each views similarity to these random anchors, that the authors claim provides alternative feature information that would otherwise be ignored. The method is evaluated on the full range of standard self-supervised benchmarks, comparing against most expected methodologies and achieving good performance improvements across the board.

**Strengths:**

**Simple approach.**
- The proposed approach is seemingly effective and simple from my understanding, presenting an interesting alternative for increasing the number of anchors with a similarity scoring mechanism to choose highly related yet opposing anchors.
- The MIM is a sensible extension of the proposed method showing the applicability to addressing the full scope of the problem setting explored, and yield significant improvements as shown in ablation studies.

**Improved Empirical performance.**
- Generally, albeit not always, the empirical performance of the method is increased compared to other benchmarked approaches across all settings, tasks and benchmarks.
- I am highly satisfied in the empirical evaluations, the authors have done a good job in including most standard benchmarks, both in terms of datasets and tasks.
- All results are presented clearly, with mostly good rigour given to the empirical analysis following standard evaluation procedures presented in comparing works.

**Strong ablations.**
- For the most part, the authors do a good job performing ablations and sensitivity analyses of the proposed method. While there are some important and missing ablations that I believe will strengthen the work, the majority of analyses are performed and discussed adequately.

**Weaknesses:**

**Text Clarity.**
- The clarity of the writing is relative poor, much of the explanations are overly complex and could be shortened. For example the paragraph from line 200. This makes the work feel bloated, with the majority of text being filler. Pages 2 - 4 could be made more concise and halved with little to no impact on the narrative.
- Following this, the justification of the method is formulated in-terms of semantic features, which although seeing the authors rationale, I feel this is somewhat misleading or obscure as no evidence is provided to back up these claims or hypothesis. There is no guarantee that the procedure captures these other features of interest as a whole, rather additional similar “intermediate or in-between” anchors (for the lack of a better term) are introduced. Please correct me if my I am misunderstanding this.
- Figure 2 needs further annotations, or description, it is not clear to the reader what it is conveying, notably the middle section.
- You mention in the abstract that your method achieves sota however, this should be clarified as being sota under this objective function formulation. There exists other SSL methods with greater performance.

**Missing details and reproducibility instructions.**
- There needs to be a more concrete description of how E is produced, updated, etc. Furthermore, details such as hyperparamterisation are missing, what random distribution are the anchors selected, this makes a huge difference on l2 normalised space? Please correct me if I’m have missed details, as in relation to my point of clarity some of these details may be lost/misunderstood.
- Key details are missing from the results section, for example how many anchors are chosen for Table 1?

**Fair comparison to other approaches with increasing anchors or prototypes targets.**
- How does your method compare to the DINO or MSN method when the number of class anchors you sample is the same as the number of prototypes? For example your method could be seen as being somewhat similar to these methods, however, I understand you are simply using a memory bank akin to MoCo to sample more anchor classes and scale the probability distribution via similarity of other judges. My question is however, is your method more performant simply because you increase the number of anchors, this can be tested by directly comparing the anchors you use to the number of classes or prototypes in these other methods (DINO or MSN).
- This above comparison would show if the performance gains are attributed to your novel selection and scaling process or simply by increasing the number of anchor points.

**Justification of shared features.**
- While I generally agree with the justification made regarding shared features, this is simply a product of the augmentation strategy involving crops. You had identified this in your conjecture, therefore it would be nice to see some additional analysis with varying crops, from your justification, you should observe a smaller delta in performance gain as the crops increase in size. As such you are proposing a method to address a self-introduced issue. I understand in practice this is not the only reason, but your justification and narrative is built on this observation.

**Minor:**
- Less of a weakness, rather a suggestion. Figure 1 could be expanded to include other results, currently it tells part of the story and given the other results are also positive it would be worth showing top-1 performance also to fairly compare agains the most common benchmark metric.
- The narrative of the introduction could be improved, for example, why does the first paragraph not address the problem statement, while the second paragraph’s justification is not as compelling as the ones presented later on. Arguably the justification in the introduction is a purposeful design choice of SSL to ensure generalised representations are learnt, while I agree with the sentiment of the authors, this section could be improved to ensure the justification is consistent.
- The motivation for MIM could be extended with a more rigorous justification.
- Weaknesses and limitations of the work are not discussed in detail, at the very least computational performance should be addressed.

**Questions:**

1. Is the size of E defined as all features of all images from the previous epoch, or is it a subset of the full dataset features? If the latter what impact does the size of E play?
2. What is the random distribution used to select the anchors? Is its uniform to ensure maximal information is preserved on the sphere?
3. Have you experimented with other strategies for selecting the anchors?
4. What is the convergence of such methods, the selection process would imply that convergence would take longer as initial bad representation anchors early in training would negatively impact this stage of training?
5. From the above question, in practice does this occur?
- In addition, I have inherently implied questions in the above weaknesses.

---

> ### Author Response · Authors · 2024-11-23
> **Clarity of writing**
>
> > The clarity of the writing is relatively poor, much of the explanations are overly complex and could be shortened. For example the paragraph from line 200. This makes the work feel bloated, with the majority of text being filler. Pages 2 - 4 could be made more concise and halved with little to no impact on the narrative.
>
> We appreciate the feedback and agree that we could have done a better job on the passages referenced by the reviewer. We are conducting a review of the paper and it should be much better (easy to read) after the changes.

---

> > ### Author Response · Authors · 2024-11-23
> > **Justification of the method and claims**
> >
> > > Following this, the justification of the method is formulated in-terms of semantic features, which although seeing the authors rationale, I feel this is somewhat misleading or obscure as no evidence is provided to back up these claims or hypothesis.
> >
> > We agree with the reviewer that further analysis would strengthen the claims we make in the paper. As mentioned by the reviewer, we hypothesize that using extra anchors to represent a hidden concept in the data, produces the effect of increasing the feature set of a concept, because the extra anchors (judges) bring in additional features that are not present or strengthened in a single anchor or prototype.
> >
> > We call the reviewer's attention to the fact that the claims are backed up by the experimentation presented in the paper. As shown in sec 3.0, the strategy produces a stable and effective algorithm to learn representations from unlabeled data. Our evaluation protocol includes several different and complementary analyses ranging from dense prediction evaluations such as object detection, instance and semantic segmentation of video (Tables 2 and 4), classic linear probing (Table 1), transfer learning (Table 3), a range of retrieval-based tasks (Tables 1 and 5), and robustness analysis (Table 6). In the majority of the results, SOVE either outperforms competitors by large margins or by small gains.
> >
> > As the reviewer points out,
> >
> > > There is no guarantee that the procedure captures these other features of interest as a whole, rather additional similar “intermediate or in-between” anchors (for the lack of a better term) are introduced. Please correct me if my I am misunderstanding this.
> >
> > While there is no theoretical guarantee that the additional features will be optimal and of interest to a particular hidden concept in the data, the reviewer captures our line of thinking. We hypothesize that these additional or intermediate features, from additional anchors, will play a positive role in the representation learning problem, which is demonstrated by the evaluation protocol, and nonetheless speaks to the effectiveness of the proposal.

---

> > > ### Author Response · Authors · 2024-11-23
> > > **Figure 2 needs further annotations**
> > >
> > > > Figure 2 needs further annotations, or description, it is not clear to the reader what it is conveying, notably the middle section.
> > >
> > > We thank the reviewer and recognize the lack of clarity in this matter. We are working on improving the descriptions and the image itself to improve the understanding of our method.
> > >
> > > In short, In Figure 2 we tried to make it an entire description of what is going on in the method with all the modules. That might be a reason why it is not so intuitive after all. In short, the figure shows the input (two augmented views of the same image) the parallel encoders, and the generated representation vector for each view. The sphere represents the container $E_c$ that holds representations from previously seen images. For each iteration, we sample a set of anchors A, the grey circles, and each anchor samples additional judges (colored rhombuses). The anchors and the selected judges are aggregated in pseudo dataset D and their labels Y. Finally, the dataset and labels are used to classify the input views consistently.

---

> ### Author Response · Authors · 2024-11-23
> **Additional descriptions**
>
> > You mention in the abstract that your method achieves sota however, this should be clarified as being sota under this objective function formulation. There exists other SSL methods with greater performance.
>
> We agree with the reviewer's assessment and will fix this issue for the final version of the paper.
>
> > There needs to be a more concrete description of how E is produced, updated, etc. Furthermore, details such as hyperparamterisation are missing, what random distribution are the anchors selected, this makes a huge difference on l2 normalised space? Please correct me if I’m have missed details, as in relation to my point of clarity some of these details may be lost/misunderstood.
>
> We apologize for the lack of additional details and we are working hard to fix these issues for the final version of the paper. We will include a detailed “Implementation Details” section containing more information regarding the dimensionalities of the various variables we refer to as well as the missing values of hyper-parameters used in practice.
>
> E is a simple container that holds representation vectors from previous iterations. In practice, $E_c$ and $E_p$ have sizes 65536 and 8192 respectively. It is updated following a FIFO (first in first out) strategy and anchors are selected using a uniform distribution.
>
> > Key details are missing from the results section, for example how many anchors are chosen for Table 1?
>
> For the main experiments, the number of anchors (size of the set A) is 8192. And each anchor selects additional k=9 judges to represent its concept  (hidden class).

---

> ### Author Response · Authors · 2024-11-23
> **Importance of additional judges and comparisons with DINO**
>
> > How does your method compare to the DINO or MSN method when the number of class anchors you sample is the same as the number of prototypes? For example your method could be seen as being somewhat similar to these methods, however, I understand you are simply using a memory bank akin to MoCo to sample more anchor classes and scale the probability distribution via similarity of other judges. My question is however, is your method more performant simply because you increase the number of anchors, this can be tested by directly comparing the anchors you use to the number of classes or prototypes in these other methods (DINO or MSN).
>
> We agree with the assessment made by the reviewer about our method. Regarding the number of anchors and their importance in the representation learning problem described in the paper, in Table 7, we assess the effectiveness of having additional judges on each pretext task. We can see that the number of judges plays an important role in extra performance. When we increase the number of additional judges per concept, from 1 (only anchor) to 9 (anchor + 8 neighbors), we can see that the kNN performance also increases. Moreover, the extra judges seem to benefit the CLS-based loss function (sec 2.2.1) more than the patch-level loss function (sec 2.2.2). To put it in perspective, using only one anchor per concept, SOVE achieves a kNN score of 69.2 while DINO kNN performance is 67.9. Using 9 additional judges boosts kNN performance to 70.2. This experiment, (Table 7) suggests that adding more judges is beneficial and that is one of the key motivations and propositions of our work.

---

> > ### Author Response · Authors · 2024-11-23
> > **Shared features**
> >
> > > While I generally agree with the justification made regarding shared features, this is simply a product of the augmentation strategy involving crops. You had identified this in your conjecture, therefore it would be nice to see some additional analysis with varying crops, from your justification, you should observe a smaller delta in performance gain as the crops increase in size. As such you are proposing a method to address a self-introduced issue. I understand in practice this is not the only reason, but your justification and narrative is built on this observation.
> >
> > We agree with the reviewer that additional experiments would make a stronger case for the claims made in the paper. Though we did not provide additional experiments with crops and augmentations as suggested, we refer the reviewer to past work where similar experiments have been conducted. As per our intuitive understanding, the views need to share important features to allow the neural network to learn useful features. Note, however, that the views cannot be equal because that would make the network optimize for a trivial solution and cannot be too different from each other.
> >
> > SimCLR [1] and BYOL [2] have studied the effect of augmentations in SSL methods that use views as a source of supervision as SOVE does. They provide evidence that views in SSL must share some features to learn useful representations. Their experimentation shows that when we gradually increase the augmentation (crop, pixel transformations, etc) strength used to create the views, the number of shared features tends to decrease and performance usually goes up. However, at some point, if the augmentation is too strong (few important shared features) the performance goes down drastically. These results align with our motivation and choices for proposing SOVE.
> >
> > Also, our method is one in a family of methods that learn representations with views. Therefore, we do not view it as trying to fix a “self-introduced” issue. Instead, we are attempting to optimize an existing family of algorithms by alleviating some of their limitations.

---

> > > ### Author Response · Authors · 2024-11-23
> > >
> > > > Less of a weakness, rather a suggestion. Figure 1 could be expanded to include other results, currently it tells part of the story and given the other results are also positive it would be worth showing top-1 performance also to fairly compare agains the most common benchmark metric.
> > >
> > > We appreciate the feedback and will consider adding additional benchmarks to Figure 1 as suggested.
> > >
> > > > The narrative of the introduction could be improved, for example, why does the first paragraph not address the problem statement, while the second paragraph’s justification is not as compelling as the ones presented later on. Arguably the justification in the introduction is a purposeful design choice of SSL to ensure generalised representations are learnt, while I agree with the sentiment of the authors, this section could be improved to ensure the justification is consistent.
> > >
> > > We thank the reviewer for the insightful suggestions and will attempt to improve the referred sections.
> > >
> > > > The motivation for MIM could be extended with a more rigorous justification.
> > >
> > > We agree with the reviewer's assessment. Due to size constraints we always need to choose what to put in the main text. We will try to introduce these relevant details in an improved version.
> > >
> > > > Weaknesses and limitations of the work are not discussed in detail, at the very least computational performance should be addressed.
> > >
> > > We appreciate and agree with the reviewer's assessment. Regarding performance, our practical experiments suggest no noticeable difference in time and memory between SOVE and iBOT. iBOT and DINO learn prototypes, hence requiring more computing and memory to update/store gradients in the backward pass. On the other hand, SOVE stores 256-d representations in two separate containers $E_c$ and $E_p$ of sizes 65536 and 8192 and does not require the extra gradients and updates of the prototypes. The containers in SOVE are updated using a regular FIFO strategy which makes it very efficient. We will incorporate these and more details in a new  “Implementation Details” section.
> > >
> > >
> > > > Is the size of E defined as all features of all images from the previous epoch, or is it a subset of the full dataset features? If the latter what impact does the size of E play?
> > >
> > > $E$ holds only a small subset of all features. In practice, $E_c$ and $E_p$ have sizes 65536 and 8192 respectively. We experimented with sizes of 32768 and 65536 for $E_c$ and found that the latter yields slightly better results.
> > >
> > >
> > >
> > > > What is the random distribution used to select the anchors? Is its uniform to ensure maximal information is preserved on the sphere?
> > >
> > > Yes, we use a uniform distribution to sample the anchors.
> > >
> > > > Have you experimented with other strategies for selecting the anchors?
> > >
> > > No, we only tested with a uniform distribution.
> > >
> > > > What is the convergence of such methods, the selection process would imply that convergence would take longer as initial bad representation anchors early in training would negatively impact this stage of training?
> > >
> > > In practice, we have observed on-par convergence when compared to existing solutions such as iBOT. With 100 epochs of pre-training a ViT-S-based SOVE, it achieves 71.6% kNN top-1 accuracy while iBOT archives 71.5% with the same 100 epochs of pre-training. However, SOVE has a booted performance gain as more epochs are used for pre-training.

---

> > > > ### Comment · Reviewer_S7GZ · 2024-11-24
> > > > **Response to Rebuttal**
> > > >
> > > > I thank the authors for providing a detailed clarification on all of my raised questions and weaknesses.
> > > >
> > > > I have replied to individual rebuttal points above when a discussion is needed, if left blank the authors have addressed my concerns adequately.
> > > >
> > > > I have faith in the authors to amend their paper with the identified points, I would stress that the authors provide a revised version which improves the clarity of writing (as addressed by all reviewers) in reasonable time such that further feedback can be made. Additionally, any revision should also include the above rebuttal points, as these strengthen the paper.
> > > >
> > > > A revised version with these corrections made will improve the quality of the work and would result in an improved score from myself.

---

> > ### Comment · Reviewer_S7GZ · 2024-11-24
> > **Response to rebuttal: Comparison to increased anchors.**
> >
> > I agree with the authors statement here, can you further clarify one further point? is this 9 judges per anchor, if so this is a significant increase in points.
> >
> > Therefore to perform a fair and a simple comparison an ablation to compare against a number of anchors in DINO or MSN to the proposed method with equal number of (anchors + judges).
> >
> > It would make sense that if you increase the number of DINO anchors to that of the proposed method (anchors + judges) you would improve performance. This is seen throughout the literature that as you increase the number anchors in the prototype methods, you naturally capture more fine-grained semantic concepts that cluster accordingly to the parent semantic concepts. Therefore, increasing the anchors should have the same impact in performance. If this does not hold, this would provide strong empirical evidence for your work.

---

> ### Author Response · Authors · 2024-11-24
> **Comparison to increased anchors**
>
> Yes, in practice, we sample A=8192 anchors, and each anchor samples an additional 8 judges, for a total of 9 judges per concept.
>
> As the reviewer suggested, increasing the number of prototypes in iBOT and DINO usually increases performance, but only up to a certain point. This behavior has been reported in iBOT[1] and DINO[2] papers.
>
> In iBOT, the optimal number of prototypes is 8192 prototypes. Going to 16384 or 4096 decreased performance, refer to page 20 iBOT paper.
>
> In DINO, the optimal number of prototypes is 65536. Going up to 262144 or down decreases performance, refer to page 16 DINO paper.
>
> As the reviewer suggested, this provides strong empirical evidence towards the efficacy or our proposed method.
>
> We hypothesize that our method performs well with more anchors/judges because each judge contributes with different (important) features to better represent the concept (cluster). While simply increasing the number of prototypes in DINO or iBOT does not have a positive effect because each prototype is optimized to represent a distinct concept.
>
> [1]- https://arxiv.org/pdf/2111.07832
>
> [2]- https://arxiv.org/pdf/2104.14294

---

### Official Review · Reviewer_raeg · 2024-11-03

**Soundness:** 2
**Presentation:** 1
**Contribution:** 2
**Rating:** 5
**Confidence:** 2

**Summary:**

The paper presents are non-parametric representation learning framework, SOVE.  This framework is aiming to ameliorate a key issue with SOTA techniques which use data augmentation to generate multiple views of the same piece of data (image).   The problem is that often this data-augmentation forces too much invariance, e.g. forcing two non-overlapping crops of the same image to be close together, even if, visually they are not, which makes the model focus on limited/irrelevant set of features.  The proposed solution is to increase the number of "judges" so that each image would have multiple shots at being assigned to a given "concept". Additionally, a non-parametric version of Masked Image Modeling is employed (perhaps reminiscing of Non-Local Means?).   The experimental results show modest improvements over SOTA methods across several standard tasks.

**Strengths:**

1. The use of hand-designed data-augmentations have long been considered suspect, exactly for the reasons outlined in the paper.  The solution proposed here -- don't try to force all views into the same representation -- makes a lot of sense, and indeed goes all the way back to the cognitive science, e.g. Eleanor Rosch work on porotype theory, exemplar theory, etc.   It also nicely connects to some early non-parametric unsupervised learning work of Tomasz Malisiewicz, especially:
Visual Memex: http://www.cs.cmu.edu/~tmalisie/projects/nips09/
and ExemplarSVM: https://www.cs.cmu.edu/~tmalisie/projects/iccv11/
as well as the Dosovitsky's ExemplarCNN: https://arxiv.org/abs/1406.6909
I suspect that connecting the present work to these older efforts would make the paper easier to understand/situate.

2.  The non-parametric MIM seems very reasonable.  Again, it would be good to understand how related it is to Non-Local Means.

3. The improvements over SOTA are modest but, given the new method is quite different, I think that is fine.

**Weaknesses:**

My main issue with the paper (and the reason I might have gotten it quite wrong?), is that it is very difficult to read:
1.  What are "concepts", "porotypes", "anchors", and "judges".  They are not clearly defined.   Especially difficult is the definition of "concept" -- in a self-supervised method it can't really be semantic (where would the semantics come from?), and yet, in the paper it's talk about as if it is semantic.
2.  The paper talks about "many non-parametric judges", but then it seems like there are only two judges for every anchor, so the number of judges is O(K), not O(N).   The definition of non-parametric model is a model which scales with data size N, whereas here it scales with number of anchors K.
3. Th "illustrative example" in Sec 2.0 is a good story, but is it actually true?   It sounds intuitively reasonable, but I don't think this is enough -- there should be some supporting evidence / experiments / visualizations on real data.   Indeed, it's possible that story is not true after all, we know that many instances of the same class in, say, ImageNet, look nothing like each other, and yet the network does not actually "discard" any features.   Indeed, networks are happy to learn even random labelings: https://arxiv.org/abs/1611.03530
Also, I really don't understand Figure 2 -- I think it needs to be explained more carefully as there is a lot happening there.
4.  Bolding convention in Table 1 is unclear.  Somehow only SOVE is bolded, even when other methods perform as well or better.

**Questions:**

1.  I might be misunderstanding something, but wouldn't a good baseline be to just increase the number of anchors (say, by 3x, so it's the same as anchors+judges)?
2.  Since the main problem this paper is trying to solve is the fact that multiple views of the same image might not have anything to do with each other, why not get rid of multiple views all together?   Instead, try to align any image to any other image -- if it works, great, if it doesn't too bad.  This seems like a more principled way to advance a non-parametric story?  Indeed, maybe just the non-parametric MIM might already be performing well enough?

My score and confidence are pretty low right now since I feel like I might be missing something key.  Looking forward to explanation from the authors.

---

> ### Author Response · Authors · 2024-11-23
> **Definitions**
>
> > What are "concepts", "porotypes", "anchors", and "judges". They are not clearly defined. Especially difficult is the definition of "concept" -- in a self-supervised method it can't really be semantic (where would the semantics come from?), and yet, in the paper, it's talk about as if it is semantic.
>
> We apologize for the lack of clarity of such definitions and will improve the descriptions of each definition in the final version of the paper. In short, we use the word “concept” as a generalized idea of a cluster - images that share a subset of features. The words “prototype” and “anchor” are used interchangeably to reference a central embedding that represents the concept (cluster), it is akin to a centroid in a regular cluster algorithm. Finally, the word “judge” refers to the action or role that each additional anchor has when voting the membership of views to a concept. We chose the word “judge” to symbolize the idea of voting, that is, each judge votes for the membership of views to a concept.
>
> Again, we apologize for the lack of clarity, we will make sure these definitions are clear to the reader in the final version. Thanks for the feedback.

---

> > ### Author Response · Authors · 2024-11-23
> > **Additional judges and definitions**
> >
> > > The paper talks about "many non-parametric judges", but then it seems like there are only two judges for every anchor, so the number of judges is O(K), not O(N). The definition of non-parametric model is a model which scales with data size N, whereas here it scales with number of anchors K.
> >
> > The main motivation of SOVE is to propose a different approach that complements prototypical methods like DINO and iBOT. These two learn a single prototype for each concept (hidden cluster). SOVE does not learn prototypes. Instead, it samples many additional prototypes (judges) that effectively increase the features of each hidden cluster (concept). SOVE keeps two containers $E_c$ and $E_p$ to store class and patch-level representations.
> >
> > In the paper, we present two non-parametric loss functions in sections 2.2.1 and 2.2.2 respectively. For each loss, we first randomly select A=8192 anchors from $E$. Then, each anchor further selects k additional judges for the concept it represents. In practice, we use k=9 for the CLS task. As the reviewers point out, the size of the pseudo dataset D grows as a factor of k (number of additional judges to be selected), O(k).
> >
> > Regarding definition, we use the term non-parametric to emphasize the fact that we do not learn the prototypes as some other methods do. Instead, we use the embeddings that are learned in every step to create a self-supervised task that effectively learns invariant representations for different views of an image.

---

> ### Author Response · Authors · 2024-11-23
>
> > The "illustrative example" in Sec 2.0 is a good story, but is it actually true? It sounds intuitively reasonable, but I don't think this is enough -- there should be some supporting evidence / experiments / visualizations on real data. Indeed, it's possible that story is not true after all, we know that many instances of the same class in, say, ImageNet, look nothing like each other, and yet the network does not actually "discard" any features. Indeed, networks are happy to learn even random labelings: https://arxiv.org/abs/1611.03530 Also, I really don't understand Figure 2 -- I think it needs to be explained more carefully as there is a lot happening there.
>
> We agree with the reviewer that additional experiments would make a stronger case for an illustrative example in sec 2.0. Though we can not introduce new experiments at this point, we refer to recent work [1] on SSL which reviews SSL from an information-theoretic perspective and discusses from a feature compression point of view what might happen in SSL models that learn from views. Also, we have evidence from past work [1, 3] showing that views in SSL must share some amount of features to learn useful representations but they can not be equal otherwise they will learn an identity function. When we gradually increase the augmentation strength used to create views in SSL, the number of shared features tends to decrease and performance usually goes up. However, at some point, if the augmentation is too strong (few shared features) the performance goes down drastically.
>
> Regarding Figure 2, we tried to make it an entire description of what is going on in the method with all the pieces, there might be a reason why it is not so intuitive after all. In short, the figure shows the input (two augmented views of the same image) the parallel encoders, and the generated representation vector for each view. The sphere represents the container $E_c$ that holds representations from previously seen images. For each iteration, we sample a set of anchors A, the grey circles, and each anchor samples additional judges (colored rhombuses). The anchors and the selected judges are aggregated in dataset D and their labels Y. Finally, the dataset and labels are used to classify the input views consistently.
>
> [1] https://www.mdpi.com/1099-4300/26/3/252
>
> [2] https://arxiv.org/abs/2002.05709
>
> [3] https://arxiv.org/abs/2006.07733

---

> > ### Author Response · Authors · 2024-11-23
> > **Bolding convention and results with additional anchors.**
> >
> > >Bolding convention in Table 1 is unclear. Somehow only SOVE is bolded, even when other methods perform as well or better.
> >
> > We thank the reviewer for the insight. For reference, we do not consider the supervised DeiT method as a valid competitor and only report its performance for general comparison.
> >
> > > I might be misunderstanding something, but wouldn't a good baseline be to just increase the number of anchors (say, by 3x, so it's the same as anchors+judges)?
> >
> > In Table 7, we provide ablation results on the effect of having additional judges on each pretext task. As shown, as we increase the number of judges the overall performance tends to increase for the CLS loss function. For the patch-level loss function (sec 2.2.2) however, adding additional judges did not translate into extra performance. Nonetheless, we show that the non-parametric version of MIM is not only stable but produces good representations as shown in the object detection and segmentation evaluations in Table 2.

---

> ### Author Response · Authors · 2024-11-23
> **Motivation clarification, multiviews and non-parametric MIM**
>
> > Since the main problem this paper is trying to solve is the fact that multiple views of the same image might not have anything to do with each other, why not get rid of multiple views all together? Instead, try to align any image to any other image -- if it works, great, if it doesn't too bad. This seems like a more principled way to advance a non-parametric story? Indeed, maybe just the non-parametric MIM might already be performing well enough?
>
>
> We would argue that this paper proposes an alternative approach to learning representations using views. Methods like DINO and iBOT learn a single prototype for each hidden concept in the data from scratch. Instead of learning prototypes, we propose to use a non-parametric space to sample images that share semantic meaning and use these images as anchors to compare and contrast views of an image and force them to learn consistent features.
>
> Instead of a single prototype, we have many semantically similar images representing a concept, each additional judge contributes with different but complementary features for that concept. As a result, the concepts are better represented than if we used a single prototype as in previous work. Previous work in SSL [1,2] demonstrates that for images, augmentations are very important, indeed, if you remove augmentations altogether, the performance of the learned features will be negligible because the network may learn an identify function, a constant vector embedding for all images.
>
> Regarding the performance of the non-parametric MIM loss function, in Table 7 we show that only MIM is not enough to learn good representations. Indeed, if we train SOVE with MIM only, it achieves 16.8% accuracy in kNN (significantly higher than iBOT with 9.5%). However, when using the two proposed loss functions (sections 2.2.1 and 2.2.2) the performance goes up to 70.0% beating iBOT and DINO.
>
> [1] - https://arxiv.org/abs/2002.05709
>
> [2] - https://arxiv.org/abs/2006.07733

---

### Official Review · Reviewer_okZU · 2024-11-04

**Soundness:** 2
**Presentation:** 1
**Contribution:** 2
**Rating:** 5
**Confidence:** 4

**Summary:**

The paper presents Self-Organizing Visual Embeddings (SOVE), a non-parametric loss design for self-supervised representation learning. To be specific, SOVE explicitly assumes multiple anchors that contribute to representing a single concept and aggregate them to represent a single soft label for self-supervised learning. Moreover, it introduces a non-parametric design of masked image modeling loss for joint usage in training. Throughout extensive experiments, the proposed method achieves superior performances across various downstream tasks, including transfer learning, image retrieval, object detection, and segmentation.

**Strengths:**

- Extensive experiments show that SOVE consistently outperforms baselines across various benchmarks, including image classification, retrieval, and segmentation tasks. This broad evaluation shows its robustness and versatility across both global and dense prediction tasks.

- Strong motivation. The authors claim the limitation of the previous self-supervised learning methodologies that share similar structures.

- Non-parametric versions of self-supervised clustering and masked image modeling loss designs are valuable.

**Weaknesses:**

- Overall, writing is uncomfortable to read and many details missing. For example, there is no notation for the dimensionality of the spaces $A, E, D, Y$, and corresponding vectors. What is the size of $E_c$ and $A$?

- The motivation of additional judges is unclear. Unlike the authors claimed $\textit{Existing solutions employ a single judge (usually learnable) to decide upon the membership of a view to a concept}$, DINO, specifically, leverages a soft-label from K prototypes. One key difference between DINO and SOVE (without MIM) is that SOVE explicitly assigns multiple prototypes for each target concept and aggregates their labels using $Y$. On this line, I am not convinced the proposed method is much more effective or superior to DINO. Table 7 also shows the close performances of DINO and SOVE without MIM (even worse than IBOT). Furthermore, it is also unclear whether the proposed non-parametric approach is actually effective. I suspect if the authors do a variant of DINO following SOVE (i.e. $E$ are learnable prototypes with enough numbers considering the size of $E$ in SOVE), the result also would be comparable.

- As shown in Table 7, MIM is a key ingredient of SOVE. However, the authors do not compare SOVE with DINOv2 which is the state-of-the-art SSL baseline using both clustering and MIM. I recommend to include a comparison with DINOv2.

- There are no full ImageNet fine-tuning comparisons.

- There are no explanations about the trade-off between the parametric and non-parametric approaches. How big is the proposed feature set and computationally expensive compared to parametric approaches like DINO?

**Questions:**

Please see the weakness section.

---

> ### Author Response · Authors · 2024-11-23
> **Clarification for variable dimentions.**
>
> > Overall, writing is uncomfortable to read and many details missing. For example, there is no notation for the dimensionality of the spaces A,E,D,Y, and corresponding vectors. What is the size of and E_c and A?
>
> We apologize for the inconsistent terms and lack of clarity regarding the dimensionality of each defined variable in the paper. We have fixed this issue and introduced a detailed “Implementation Details” section containing all the relevant information required by the reviewer.
>
> Here we describe the dimensions of each variable, as requested.
>
> As mentioned in section 2.2.2 of the paper,
> > “D can be viewed as a dataset containing K pseudoclasses, each containing k + 1 observations, i.e., anchors ai, and their k nearest neighbors.”
>
> In practice, we sample K=8192 anchors (set A) and each anchor selects k=8 additional neighbors or judges. Note that anchors and additional judges are selected from the containers E. This gives a total of 8192 anchors + 8*8192=65536, neighbors which sums up to 73728 judges (9 for each concept), 256-d visual embeddings. Y are the pseudo-labels associated with D, you can view them as one-hot vectors.
>
> The letter E is used as a generalized notation for the containers $E_c$ and $E_p$. Each container holds representations from the [CLS] and patch-level tokens respectively. In practice, $E_c$ and $E_p$ have sizes 65536 and 8192 respectively.

---

> > ### Author Response · Authors · 2024-11-23
> > **Motivation, Comparison to DINO and performance.**
> >
> > > The motivation of additional judges is unclear. Unlike the authors claimed  ""exisitng solutions employ a single judge (usually learnable) to decide upon the membership of a view to a concept"", DINO, specifically, leverages a soft-label from K prototypes. One key difference between DINO and SOVE (without MIM) is that SOVE explicitly assigns multiple prototypes for each target concept and aggregates their labels using Y.
> >  On this line, I am not convinced the proposed method is much more effective or superior to DINO. Table 7 also shows the close performances of DINO and SOVE without MIM (even worse than IBOT). Furthermore, it is  also unclear whether the proposed non-parametric approach is actually effective. I suspect if the authors do a variant of DINO following SOVE (i.e., E  are learnable prototypes with enough numbers considering the size of E in SOVE), the result also would be comparable.
> >
> > The main motivation of SOVE is to propose a different approach that complements prototypical methods like DINO which leans a single prototype for each hidden concept. As the reviewer points out, SOVE does not learn prototypes like DINO and it samples many additional prototypes (judges) that effectively increase the concept features. Also, we would like to call the reviewer's attention to the discrepancy in performance between SOVE and DINO. In Table 7, SOVE outperforms Dino by +0.8 when MIM is not employed. With MIM, the improvement is +2.1 and it surpasses iBOT by +0.9 not otherwise. Besides, SOVE shows that a non-parametric version of MIM not only works but delivers slight improvements over the learnable version from iBOT. In addition, SOVE outperforms both iBOT and DINO on a variety of other downstream tasks such as object detection and instance segmentation (Table 2), transfer learning by fine-tuning (Table 3), video and object segmentation (Table 4) and image retrieval (Table 5). We strongly believe such numbers demonstrate SOVE's ability to learn good representations and its superiority against DINO is well demonstrated.

---

> > > ### Author Response · Authors · 2024-11-23
> > > **non-parametric MIM and Dinov2**
> > >
> > > > As shown in Table 7, MIM is a key ingredient of SOVE. However, the authors do not compare SOVE with DINOv2 which is the state-of-the-art SSL baseline using both clustering and MIM. I recommend to include a comparison with DINOv2.
> > >
> > > We agree with the reviewer that MIM is a key ingredient in SOVE. However, we would like to point out that it is not a regular MIM implementation as presented and used by iBOT and DINOv2, instead, we propose and demonstrate a new approach that works in the non-parametric space, and call it the non-parametric MIM task. As per DINOv2, we agree it is the current state of the art, we emphasize however that DINOv2 is trained in a much larger transformer using a collection of datasets much larger than the ImageNet dataset used in this paper, and that DINOv2 results for the “smaller” vit-s/b/l architectures are achieved via model distillation. As a result, we believe it is not a fair comparison with SOVE which trains solely on ImageNet-1k.

---

> > > > ### Author Response · Authors · 2024-11-23
> > > > **ImageNet fine-tuning comparisons**
> > > >
> > > > > There are no full ImageNet fine-tuning comparisons
> > > >
> > > > We agree with the reviewer about the lack of ImageNet fine-tuning evaluations. We would like to point out that we can only do so much and it is impractical to cover all the spectrum of evaluation techniques used for self-supervised learning. For ImageNet, we chose Liner probing, and k-NN (Table 1) mostly because they assess the learned representation as is and we believe are more rigorous protocols since they do not change the representation. Moreover, we believe that one of the benefits of learning good representations is that they should work out of the box and prevent the user from spending additional time and resources fine-tuning large collections of data. For this reason, we also provide semi-supervised fine-tuning in Table 1. Also, we provide fine-tuning experiments on different datasets such as iNaturalist, Followers, and Cars. We believe this protocol is much more computationally friendly and rigorous since evaluation is done on a different modality than the one used for pre-training.

---

> > > > > ### Author Response · Authors · 2024-11-23
> > > > > **Trade-off between the parametric and non-parametric approaches.**
> > > > >
> > > > > > There are no explanations about the trade-off between the parametric and non-parametric approaches. How big is the proposed feature set and computationally expensive compared to parametric approaches like DINO?
> > > > >
> > > > > We recognize the lack of a trade-off between the parametric and proposed non-parametric approach. We are working on such evaluations and will be able to provide further insights in the final version of the manuscript.
> > > > >
> > > > > Our results so far suggest no noticeable difference in time and memory between SOVE and iBOT/DINO. iBOT and DINO learn prototypes, hence require more computing and memory to compute/store gradients for the backward pass. On the other hand, SOVE stores 256-d representations in two separate containers $E_c$ and $E_p$ of sizes 65536 and 8192 and does not require the extra gradients and updates of the prototypes. The containers in SOVE are updated using a regular FIFO strategy which makes it very efficient.

---

> ### Comment · Reviewer_okZU · 2024-11-27
>
> Thank you for your reply. However, some of your responses still leave me uncertain. It has been well-established in the DINOv2 paper that DINO is more effective when combined with MIM. Therefore, to confirm the unique effectiveness of the proposed paper, it should be rigorously compared with DINO without MIM or with DINOv2 incorporating MIM. However, the author does not appear to provide any new or meaningful evidence in this regard. Furthermore, behavioral observation in the ImageNet fine-tuning domain is very important in the field of self-supervised learning. For example, MAE demonstrates weak linear probing and k-NN performance, but it outperforms many models when fine-tuned. Finally, there is a lack of numerical evidence regarding the trade-off between parametric and non-parametric approaches. I will maintain my score as is.

---

> ### Author Response · Authors · 2024-11-27
> **Further clarifications for DINO and DINOv2**
>
> We appreciate the reviewer's feedback and would like to address their concerns as follows:
>
> > It has been well-established in the DINOv2 paper that DINO is more effective when combined with MIM. Therefore, to confirm the unique effectiveness of the proposed paper, it should be rigorously compared with DINO without MIM or with DINOv2 incorporating MIM.
>
> We agree that the MIM task is crucial for our method, as shown in Table 7, as well as in DINOv2 and iBOT. However, we would like to highlight the following points:
>
> > it should be rigorously compared with DINO without MIM [...]
>
> First, it is important to note that DINO does not use MIM in its original implementation.
> To the reviewer's point, we have rigorously compared our method against "DINO without MIM" in the following tables:
>
> - Table 1- Linear probing, semi-supervised finetuning, and k-NN evaluations.
> - Table 2- Object detection and instance segmentation on COCO and semantic segmentation on ADE20k.
> - Table 3- Transfer learning by fine-tuning SSL methods on smaller datasets.
> - Table 4 - Video object segmentation on DAVIS 2017.
> - Table 5 - Image retrieval.
> - Table 7 - Parametric vs non-parametric tokenizers.
>
> Alternatively, the reviewer asks for:
> > it should be rigorously compared [...] with DINOv2 incorporating MIM.
>
> As stated in the DINOv2 paper [1], DINOv2 is essentially a version of iBOT scaled to a large dataset and ViT architecture.
> In fact, iBOT can be considered "DINO with MIM", since the MIM task is the main difference between iBOT and DINO.
> Therefore, comparing against iBOT matches exactly the reviewer's request.
>
> Since direct comparisons with DINOv2 are unfair, for reasons already stated above, we compare against iBOT on all 7 Tables presented in the paper, on different benchmarks such as object detection, segmentation, image retrieval, semi-supervised learning, and more.
>
> > Furthermore, behavioral observation in the ImageNet fine-tuning domain is very important in the field of self-supervised learning. For example, MAE demonstrates weak linear probing and k-NN performance, but it outperforms many models when fine-tuned.
>
> We agree with the reviewer that "behavioral observation in the ImageNet fine-tuning domain is important.". However, as we explained, we can only do so much regarding experiments, and fine-tuning on the full ImageNet is costly and time-consuming. That is why we provided fine-tuning on ImageNet **in low data regime** using 1% and 10% of ImageNet labels - Table 1.
>
> Moreover, we are not trying to dismiss the importance of such a benchmark. Instead, we believe that many important applications in SSL require out-of-the-box representations and that is our main focus.
>
> As the reviewer pointed out with MSN, and we agree, some methods perform better in one benchmark (fine-tuning vs out-of-the-box) than the other.
> Our line of thinking however is that, after pre-training for so many hours and so much data, one should expect the out-of-the-box representations to be useful, and not have to spend a more considerable amount of resources to fine-tune on a large annotated dataset to obtain good results.
>
> > Finally, there is a lack of numerical evidence regarding the trade-off between parametric and non-parametric approaches.
>
> We are working on measuring the trade-off performance between parametric and non-parametric methods as requested. We will attempt to do so before the hard time limit to upload a new version of the paper. Otherwise, we will provide the evidence here and update the paper for a final version.
>
> We hope these additional points solve the reviewer's questions about our work.

---

> > ### Author Response · Authors · 2024-11-27
> > **Numerical evidence for parametric and non-parametric trade-off.**
> >
> > Please, find the new version of the paper, which now includes a section (Appendix B1) on the trade-off between parametric and non-parametric methods.
> >
> > Thanks to previous work [1], we were able to quickly provide the numerical evidence requested by the reviewer.
> >
> > In short, SOVE requires a similar training time and memory footprint as iBOT. The main differences between the two are the parametric vs. non-parametric approach and the judge selection method proposed by SOVE.
> >
> > We hope this numerical evidence addresses the reviewer's uncertainties about our work.
> >
> > [1]- https://arxiv.org/html/2407.17486v1

---

> ### Author Response · Authors · 2024-12-01
> **Results for Fine-tuning on ImageNet-1M**
>
> After running the experiments, we report results for fine-tuning SOVE pre-trained encoders on the ImageNet dataset.
> We followed the protocol from [1] and fine-tuned a 400ep ViT-base pre-trained SOVE encoder on the ImagaNet-1M for 100 epochs.
> The Table below is an extension of Table 1 of the paper and reports top-1 accuracy for the fine-tuning experiment on the last column (FT).
> SOVE outperforms, DINO and iBOT by +0.6 and +0.2 respectively.
> The results support previous experiments and attest to our method's effectiveness in both: (1) retrieval and (2) fine-tuning downstream tasks.
> For reference, we also included fine-tuning results for MSN.
>
> We hope this further evidence, as requested by the reviewer, is sufficient to clear any open questions about our work.
>
> | Method | Arch.    | Ep.  | Lin.     | 1%       | 10%      | k-NN     | FT       |
> |--------|----------|------|----------|----------|----------|----------|----------|
> | DINO   | ViT-B/16 | 400  | 78.2     | 64.4     | 76.3     | 76.1     | 83.6     |
> | iBOT   | ViT-B/16 | 400  | 79.5     | 68.5     | 78.1     | 77.1     | 84.0     |
> | MSN    | ViT-B/16 | 600  |          |          |          |          | 83.4     |
> | SOVE   | ViT-B/16 | 400  | **79.9** | **69.5** | **78.2** | **78.4** | **84.2** |
>
>
> [1]- https://arxiv.org/abs/2111.07832

---

### Author Response · Authors · 2024-11-25
**New version with Implementation details**

As requested by most of the reviewers, we just submitted a new version of the paper containing an "Implementation Details" section in the appendix. In this section, we disclose the hyperparameters used to train our method and the practical shapes of variables that were missing in the original submission.

We hope this addition can clarify the questions raised by the reviewers.

---

### Author Response · Authors · 2024-11-26
**Thank you to the reviewers and revised version.**

**Thank you to the reviewers for their insightful comments.**

Below, we summarize the key strengths of our paper as identified by the reviewers. Notably, our comprehensive experiments demonstrate that our method consistently surpasses baselines across a range of benchmarks, illustrating its robustness and adaptability (R.okZU, R.S7GZ). The prevalence of self-supervised augmentation techniques underscores the strong motivation for our work, affirming its soundness (R.okZU, R.raeg). In addition to its simplicity and effectiveness (R.S7GZ), the integration of the MIM extension with our method is both practical and yields significant enhancements (R.S7GZ). The non-parametric loss we introduce adds considerable value (R.okZU). Considering the methodological distinctions from existing approaches, the modest gains reported are seen as satisfactory (R.raeg, R.S7GZ).

---

As promised in the discussion comments below, we uploaded a **new version of the paper** with many corrections, additions, and clarifications in the main text.

Below, we summarise some of the fixes requested by reviewers.

Other points have been directly addressed in the discussion comments below.

We hope these additions clarify the valid questions and inquiries raised by the reviewers.

> For example, there is no notation for the dimensionality of the spaces A,E,D,Y, and [...] is the size of and E_c and A?
**FIXED**

> How big is the proposed feature set and computationally [...]
**FIXED**

> What are "concepts", "prototypes", "anchors", and "judges". They are not clearly defined.
 **FIXED**


> Bolding convention in Table 1 is unclear. Somehow only SOVE is bolded, [...] perform as well or better.
**FIXED**


> The clarity of the writing is [...] Pages 2 - 4 could be made more concise and halved with little to no impact on the narrative.
**FIXED**


> Figure 2 needs further annotations, or description, it is not clear to the reader what it is conveying, notably the middle section.
**FIXED**

> You mention in the abstract that your method achieves sota however, this should be clarified as being sota under this [...] performance.
**FIXED**

> There needs to be a more concrete description of how E is produced, updated, etc. Furthermore, [...] of may be lost/misunderstood.
**FIXED**


> Key details are missing from the results section, for example how many anchors are chosen for Table 1?
**FIXED**


> Is the size of E defined as all features of all images from the previous epoch, or is it a subset of the full dataset features? [...] size of E play?
**FIXED**

> What is the random distribution used to select the anchors? Is its uniform to ensure maximal information is preserved on the sphere?
**FIXED**

---

### Meta-Review · Area_Chair_9kde · 2024-12-21

**Metareview:**

The paper proposes a new self-supervised learning method that improves upon existing clustering techniques by using multiple "judges" (i.e., semantically similar representations) rather than a single prototype for each concept.  It uses two novel non-parametric loss functions: non-parametric cluster assignment prediction and non-parametric Masked Image Modeling (MIM).

The paper received ratings 5,5,5,3.  Reviewers converged onto very poor writing, and they were unconvinced by either the ideas, technical details, or experimental gains.  The authors's rebuttal provided detailed clarifications, especially regarding the motivation and non-parametric MIM. They acknowledged that some aspects of the paper required more experimentation and clearer explanations. They argued that the increased number of judges (anchors) helped capture more nuanced features and improved performance. They also defended their approach by emphasizing its efficiency, especially compared to DINO and iBOT in terms of computational cost.

While the rebuttal clarified some confusion, reviewers remained negative about accepting the paper.  1) Multiple reviewers highlighted that the paper was difficult to read, with unclear definitions and explanations of key concepts.  2) The motivation behind the method was not convincingly backed up by experiments.  3) Modest performance gains were not sufficiently compelling to justify the approach's novelty, especially when compared to other state-of-the-art methods.  The authors were encouraged to revise the paper with better clarity and more rigorous experimental validation before resubmission.

**Additional Comments On Reviewer Discussion:**

Two reviewers engaged with the authors on their detailed rebuttal. One reviewer acknowledged reading all the reviews and rebuttal.  The fourth reviewer was pinged twice for post-rebuttal actions.  Overall, the reviewers still found the paper poorly written, hard to understand, and unconvincing, and the only way to address all the reviewers' issues would be a "major revision and resubmit".

---

### Decision · Program_Chairs · 2025-01-22

Reject